# AI-Based screening for thoracic aortic aneurysms in routine breast MRI

Dimitrios Bounias [1,2], Tobit Führes[3], Luise Brock[3], Johanna Graber[3], Lorenz A. Kapsner [3,4], Andrzej Liebert [3], Hannes Schreiter [3], Jessica Eberle [3], Dominique Hadler [3], Dominika Skwierawska [3], Ralf Floca [1,5], Peter Neher [1,6,7,8], Balint Kovacs [1,2], Evelyn Wenkel[9], Sabine Ohlmeyer[3], Michael Uder[3], Klaus Maier-Hein[1,2,6,7,8,10] & Sebastian Bickelhaupt [3] ✉

Prognosis for thoracic aortic aneurysms is significantly worse for women than men, with a higher mortality rate observed among female patients. The increasing use of magnetic resonance breast imaging (MRI) offers a unique opportunity for simultaneous detection of both breast cancer and thoracic aortic aneurysms. We retrospectively validate a fully-automated artificial neural network (ANN) pipeline on 5057 breast MRI examinations from public (Duke University Hospital/EA1141 trial) and in-house (Erlangen University Hospital) data. The ANN, benchmarked against 3D-ground-truth segmentations, clinical reports, and a multireader panel, demonstrates high technical robustness (dice/clDice 0.88-0.91/0.97-0.99) across different vendors and field strengths. The ANN improves aneurysm detection rates by 3.5-fold compared with routine clinical readings, highlighting its potential to improve early diagnosis and patient outcomes. Notably, a higher odds ratio (OR = 2.29, CI: [0.55,9.61]) for thoracic aortic aneurysms is observed in women with breast cancer or breast cancer history, suggesting potential further benefits from integrated simultaneous assessment for cancer and aortic aneurysms.

With a reported incidence of about 5-10/100.000 patient years[1] and a prevalence of 0.16–0.34%[2], thoracic aortic disease manifesting as thoracic aortic aneurysm is rare compared to the more frequent abdominal aortic aneurysms[3]. Screening programs using ultrasound (US) have been implemented only for abdominal aortic aneurysms and only in men, as a screening program is considered unlikely to be cost-effective in women[4].

Thoracic aortic aneurysms are often "silent killers", with screening only offered to high-risk groups such as first-degree relatives of patients with diagnosed thoracic aortic disease[5]. An increasing incidence of thoracic aortic aneurysms in the general population has been suggested[6], though reports vary[7]. While they are rarer in women than men, women are three times more likely to develop rupture or

[1]German Cancer Research Center (DKFZ) Heidelberg, Division of Medical Image Computing, Im Neuenheimer Feld 280, Heidelberg, Germany. [2]Medical Faculty Heidelberg, Heidelberg University, Im Neuenheimer Feld 672, Heidelberg, Germany. [3]Radiological Institute, Uniklinikum Erlangen, Friedrich-Alexander-Universität Erlangen-Nürnberg (FAU), Maximiliansplatz 3, Erlangen, Germany. [4]Chair of Medical Informatics, Friedrich-Alexander-Universität Erlangen-Nürnberg (FAU), Wetterkreuz 15, Erlangen-Tennenlohe, Germany. [5]Heidelberg Institute of Radiation Oncology (HIRO), National Center for Radiation Research in Oncology (NCRO), Im Neuenheimer Feld 280, Heidelberg, Germany. [6]German Cancer Consortium (DKTK), Partner Site Heidelberg, Im Neuenheimer Feld 280, Heidelberg, Germany. [7]Pattern Analysis and Learning Group, Department of Radiation Oncology, Heidelberg University Hospital, Im Neuenheimer Feld 400, Heidelberg, Germany. [8]National Center for Tumor Diseases (NCT), Heidelberg University Hospital (UKHD) and German Cancer Research Center (DKFZ), Im Neuenheimer Feld 460, Heidelberg, Germany. [9]Radiologie München, Burgstraße 7, München, Germany. [10]Faculty of Mathematics and Computer Science, Heidelberg University, Im Neuenheimer Feld 205, Heidelberg, Germany. ✉e-mail: sebastian.bickelhaupt@uk-erlangen.de

dissection. Women are also about 40% more likely to die of a thoracic aortic aneurysm[8] and have higher hospital mortality[6] compared to men. Of special relevance, rupture and dissection seem to occur at a smaller aneurysm size in women[5]. Early detection of thoracic aortic aneurysms has been suggested to allow stratified preventive medical and lifestyle interventions[5], surveillance with follow-up imaging to monitor potential progression, and planning of interventions. Notably, emergency status has been suggested as a highly relevant predictor of death associated to surgery[9]. This highlights the need to reduce the risk of aortic dissection and rupture, especially considering that ruptures are reportedly associated with mortality rates > 90%[1,10]. Breast magnetic resonance imaging (MRI) use has been increasing by approximately 5–12% per year[11,12], covering a spectrum of clinical indications. Screening based on individual risk factors commonly includes breast MRI[13,14]. MRI has been demonstrated to offer superior sensitivity, particularly in women with dense and extremely dense breast tissue, allowing to reduce interval cancer rate[13–15] and prompting the European society of breast imaging (EUSOBI) to suggest its supplemental use in women with extremely dense breast[16]. Breast MRI examinations - regardless of indication – routinely include gadolinium-based contrast agents (GBCA) enhanced (CE) acquisitions[17,18]. Intrinsically, breast MRI provides anatomical coverage extending into the mediastinal thorax, enabling evaluation of the large thoracic vessels, depending on the chosen field of view (FOV).

Due to the substantial and growing healthcare burden[19] of thoracic aortic aneurysms and the potential for safer treatment if they are detected early, we conceptualized using routine breast MRI examinations for automated detection of thoracic aortic aneurysms through artificial neural networks (ANNs). This approach could leverage routine imaging examinations to enable effortless screening for a rare disease without requiring any additional medical appointments or procedures for the patient.

In this multi-center study, we present a scalable, fully automated approach for secondary use of breast MRI examinations for thoracic aortic aneurysm screening using an ANN that can be implemented in a clinical routine-ready software infrastructure.

## Results

Our study evaluated the developed ANN approach for background screening of thoracic aortic aneurysms in $n = 5057$ breast MRI examinations acquired at different institutions, with MRI devices covering all major MRI manufacturers and clinical field strengths for independent testing (Erlangen test dataset: $n = 3232$, DUKE: $n = 922$, and EA1141: $n = 903$). Characteristics of the different trials and analysis pathway are given in Fig. 1 and Supplementary Fig. 1, respectively, while cohort demographics and MRI acquisition techniques are given in Table 1, Supplementary Table 1, and Supplementary Table 2.

**Performance of the ANN to screen for thoracic aortic aneurysms**
**Automated ascending (AA) and descending thoracic aorta (DA) assessment and detection of aneurysms by the ANN.** Average ascending maximum aortic diameters were determined by the ANN to be 2.97 cm (CI: [2.96, 2.99]) for the Erlangen test set, 2.84 cm (CI: [2.82, 2.85]) for Duke, and 2.98 cm (CI: [2.97, 3.00]) for the EA1141 trial data. Correspondingly, the average maximum descending aortic diameter was 2.22 cm (CI: [2.21, 2.24]) for the Erlangen test set, 2.21 cm (CI: [2.20, 2.23]) for Duke, and 2.22 cm (CI: [2.21, 2.23]) for the EA1141 data. Figure 2 presents the distribution of the calculated diameters. Analytical results can be found in Table 2.

The ANN demonstrated a 0.28% flagging frequency (9/3232) for potential aortic aneurysms according to the AHA definitions within the Erlangen test dataset. Case-by-case review by the radiologist panel demonstrated, that the ANN revealed $n = 8$ examinations (0.24%) belonging to $n = 7$ patients (7/2258, patient prevalence 0.31%) with thoracic aortic aneurysms, amongst $n = 6$ in the AA (mean diameter:

48 mm, range: 45–59 mm) and $n = 2$ in the DA (mean diameter: 36 mm, range: 35–37 mm). The ANN had flagged one case due to incorrect segmentation, which was determined during panel review. For the EA1141 examinations the algorithm flagged $n = 3$ cases of potential AA aneurysms (mean diameter: 48 mm, range 46–52 mm) for $n = 2$ patients. For the Duke dataset, the ANN indicated $n = 2$ cases of aneurysms ($n = 1$ in the AA with 45 mm and $n = 1$ in the DA with 35 mm). All those Duke and EA1141 cases were verified true positive, and the ANN did not create any aneurysm flagging due to mis-segmentation or mis-classification.

Evaluating the positive predictive value (PPV) for cases which were called by the ANN in the independent datasets depending on different thresholds resulted in the following PPVs: For the AA the PPV1 at 20.8% (10/48), PPV2 at 55% (10/18) and PPV3 at 100% (10/10). For the DA, the PPV1 was 6.6% (3/45), PPV2 50% (3/6) and PPV3 75% (3/4).

Analysis of all diameter cases flagged by the ANN (for PPV1, PPV2 and PPV3) compared to manual control segmentations indicated a mean submillimeter deviation of 0.524 mm (± 0.9 mm, $p < 0.001$) for the ANN compared to the mean human diameter assessments. Using the visual independent evaluation by the radiologist panel, the NPV (at PPV1) was found to be 100% for the ANN (which demonstrated a mean variation of 0.3 mm in between the two readers, $p < 0.001$).

Analytical results are presented in Table 3. Example predictions can be found both for the Erlangen test dataset and the EA1141 and DUKE datasets in Supplementary Figs. 2 and 3.

**Automated risk analysis of the thoracic ascending aorta.** The Aortic Size Index (ASI) and the Aortic Height Index (AHI) are meant to adjust the AA diameter using the patient's Body Size Area (BSA) and height, respectively, and apply only for ascending thoracic aorta (AA) diameters $\geq 3.5$ cm. There were $n = 383$ such examinations (AA diameter $\geq 3.5$ cm) with corresponding height/weight available in the three independent test datasets (Erlangen test dataset: $n = 325$, Duke: $n = 6$, EA1141: $n = 52$). Of those, $n = 38$ examinations were classified as elevated risk with a 7% predicted annual risk of complications (AHI > 2.43). Included in those $n = 38$ examinations, only 2 patients were classified with an elevated ASI risk of 8% (ASI > 2.75), highlighting a difference between the two risk models. All $n = 10$ predicted AA aneurysms were classified as elevated risk (using AHI). There were $n = 9$ patients with AA diameter < 4.0 cm, which would typically fall below the dilatation threshold, classified as elevated risk (Fig. 3).

**Improvement of detection rate by the ANN over human clinical routine assessments**
Analysis of the clinical radiologist reports, which were established during clinical routine in the independent Erlangen test dataset revealed a clinical reporting rate for thoracic aneurysmatic aortic disease (both dilatation and aneurysm) of 0.06%, corresponding to a total of 2/3232 examinations, and a patient-level reporting prevalence of 0.09% (2/2258 patients). As of the $n = 2$ clinically reported cases, written reports were held descriptive, reporting an "extended" aorta. All those reports of thoracic AA abnormalities (ascending aorta diameter for case 1: 45 mm, case 2: 61 mm) were confirmed by the panel board as thoracic AA aneurysms.

The ANN detected cases included all human-described thoracic aneurysmatic aortic disease cases from clinical routine (2/2 patients/examinations, sensitivity of detecting clinically reported patients/cases: 100%). The additional $n = 6$ examinations of $n = 5$ patients with thoracic aortic aneurysms revealed by the ANN thus resulted in a 3.5-fold improvement of the relative detection rate (RDR) on the patient level ($n = 7$ patients with thoracic aorta aneurysms revealed by the ANN instead of $n = 2$ by clinical routine), when compared to real-world human routine readings in the $n = 2258$ patients from the Erlangen test dataset.

## Higher prevalence of thoracic aortic aneurysms in women with breast cancer or breast cancer history

In the Erlangen test set, the average maximum (of each patient) diameter of the AA was 3.04 cm in women with breast cancer and 2.96 cm in healthy patients. The DA average maximum diameter was 2.27 cm and 2.21 cm, respectively. The Mann-Whitney U test showed the difference to be significant ($p < 0.001$ for both AA and DA), and similarly, the Kolmogorov–Smirnov test showed the distributions to be significantly different ($p \leq 0.002$ for both AA and DA). Examinations of women with breast cancer presented a higher average maximum diameter and a higher odds ratio of presenting with a thoracic aortic aneurysm (OR = 2.14, CI: [0.51, 8.97]). The effect slightly increased when women with a history of breast cancer were also considered (OR = 2.29, CI: [0.55,9.61]) and slightly reduced when patients with current breast cancer were excluded and only history of breast cancer was analyzed (OR = 1.91, CI: [0.32,11.48]). The distributions are presented in Supplementary Fig. 4.

## Performance analysis of the ANN for volumetric segmentation of the thoracic aorta

**Dice coefficient and clDice.** During training, using the Erlangen training set, the ANN achieved a Dice or 0.95 and a clDice of 1.00. Applying the ANN to the independent test datasets, the agreement between thoracic aorta segmentation masks predicted by the ANN and the ground truth segmentations revealed a Dice coefficient and centerline Dice (clDice) of 0.91 (CI: [0.90, 0.92]) / 0.99 (CI: [0.98, 0.99]) for the Erlangen test dataset, 0.88 (CI: [0.87, 0.90]) / 0.97 (CI: [0.95, 0.98]) for Duke and of 0.91 (CI: [0.89, 0.92]) / 0.99 (CI: [0.99, 1.00]) for the EA1141 dataset, respectively. ANNs trained with a different number of training samples ($n = 24,48,72$) performed similarly (Supplementary Fig. 5).

**Missegmentations and error causes.** Manual assessment of all ANN-derived segmentations in the independent test datasets revealed an error rate of the ANN pipeline of 0.56% for the Erlangen test dataset,

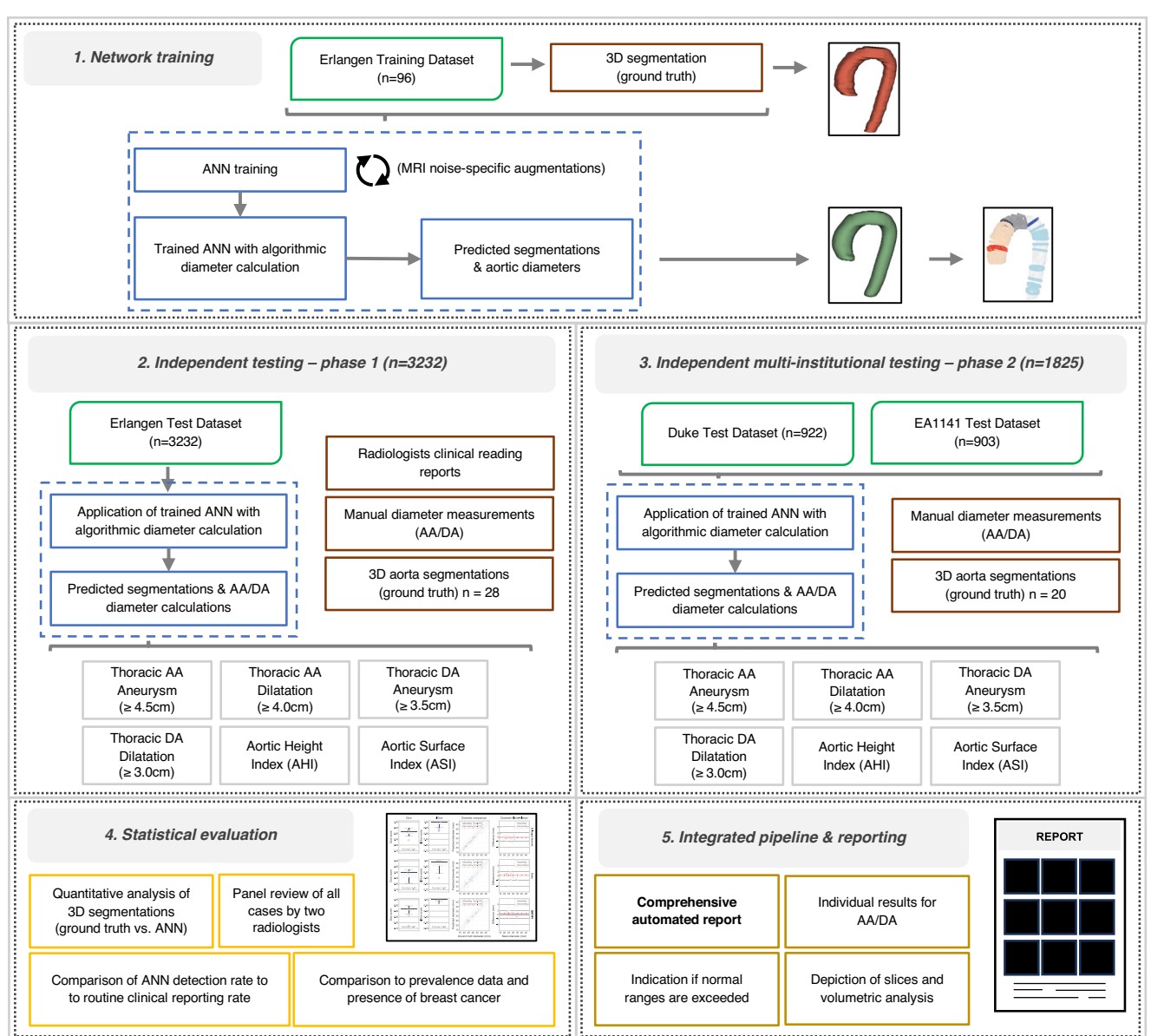

**Fig. 1 | Depiction of the study workflow indicating the steps of training, independent testing and statistical evaluation as well as the integration of the developed artificial neural network (ANN) in a pipeline creating PDF reports.** The first step conducted was training of the ANN (1. Network training), followed by independent testing in the Erlangen test dataset (2. Independent testing) and multicentric independent testing in additional public datasets (3. Independent multi-institutional testing). Results were statistically evaluated (4. Statistical evaluation) and an integrated pipeline for processing and reporting was established (5. Integrated pipeline & reporting), allowing for creating reports of the analysis. AA Ascending Aorta, DA Descending Aorta.

4.56% for the Duke dataset, and 1.55% for EA1141. Details and causes for the errors of the ANN are given in Supplementary Table 3.

**Predicted and ground-truth diameter measurements.** The agreement between the predicted and ground truth maximum diameter assessments for the thoracic ascending aorta (AA) and descending aorta (DA) revealed no significant difference using a t-test with paired samples (AA: $p = 0.08/p = 0.71/p = 0.12$ for the Erlangen test set, Duke, and EA1141 respectively and DA: $p = 0.09/p = 0.20/p = 0.10$). Figure 4 presents analytical results for segmentation performance. Further, $n = 1060$ random manual diameter measurements were performed as control (in $n = 560$ breast MRI examinations). The mean variation was 1.52 (CI: [1.45, 1.58]) mm between the ANN pipeline results and the manual measurements of the radiologists for the Erlangen test set ($n = 357$ AA measurements, $n = 358$ DA measurements, in $n = 360$ examinations), with the inter-rater mean variation being 1.61 (CI: [1.50, 1.72]) mm. For Duke ($n = 77$ AA measurements, $n = 79$ DA measurements, in $n = 100$ examinations), the mean variation was 1.52 (CI: [1.39, 1.65]) mm and the inter-rater 1.69 (CI: [1.47, 1.91]) mm. For EA1141 ($n = 98$ AA measurements, $n = 71$ DA measurements, in $n = 100$ examinations), the mean variation was 1.52 (CI: [1.32, 1.72]) mm and the inter-rater 1.72 (CI: [1.48, 1.95]) mm. Figure 5 presents analytical results for diameter measurement differences.

**Robustness on repeated breast MRI examinations.** In addition, as a secondary and indirect measure of performance for the segmentations, we calculated the diameter agreement between consecutive examinations of the same patient (Fig. 6). The assumption is that despite variations in aorta diameter being possible between different examinations of the same patient (due to natural changes that come with age, environmental and lifestyle factors, or aortic disease), the calculated diameter should not typically see extreme variations between visits. 2893/3382 (85.5%) of calculated AA/DA/arch diameter pairs were found to be within 2 mm of each other. Significance testing

was performed using a $t$ test with paired samples, with the hypothesis that there is no difference between repeat measurements of the same patient and the paired samples representing diameter measurements of two repeat examinations of the same patient. Testing indicated a difference only for the AA. Specifically, (i) for the Erlangen test set, $p = 0.011$ for AA measurements, p = 0.876 for DA, $p = 0.598$ for the aortic arch (ii) for EA1141, $p = 0.017$ for AA, $p = 0.749$ for DA, $p = 0.349$ for the aortic arch.

**Stability across different post-GBCA-administration timepoints.** Assessing the stability of the ANN segmentation process across different timepoints after GBCA injection has demonstrated a vastly unchanged DICE coefficient with the ground truth, compared to using the first timepoint (Fig. 7). Specifically, dice of 0.89 (CI: [0.89, 0.90]) for acquisitions belonging to the second timepoint post-GBCA-administration, dice of 0.89 (CI: [0.88, 0.89]) for the third timepoint, 0.88 (CI: [0.87, 0.88]) for the fourth, and 0.87 (CI: [0.86, 0.87]) for the fifth. The average absolute diameter difference was in the 0.9–1.2 mm range.

**Adequacy of the plane categorization process.** For estimating the adequacy of selecting different segments of the thoracic aorta (ascending, arch, descending), the manual evaluation of maximal distance in between the anatomic supra-aortic branches and the automatically selected section indicated a mean absolute difference of 8.8 mm and 10.8 mm for the ascending/arch and the arch/descending position.

**Additive burden of clarification examinations and associated costs.** Clinical application of the ANN will cause a varying number of false-positive findings depending on the thresholds chosen. For estimating the associated burden of patients undergoing clarification examinations it was found that for each aneurysm detected in between $n = 1$ and $n = 15$ patients might undergo additive imaging

**Table 1 | Patient demographics and acquisition characteristics**

|  | In-house dataset (training) | Erlangen test dataset | DUKE | EA1141 |
|---|---|---|---|---|
| **Breast MRI examinations** | 96 | 3232 | 922 | 903 |
| **Patients** | 96 | 2258 | 922 | 480 |
| **Average age (years)** | 47.9 (96[a]) | 49.5 (3232[a]) | 52.9 (540[a]) | 55.3 (820[a]) |
| **Average weight (kg)** | 65.2 (96[a]) | 70.1 (3232[a]) | 76.4 (810[a]) | 70.9 (825[a]) |
| **Average height (m)** | 1.66 (96[a]) | 1.67 (3232[a]) | 1.66 (164[a]) | 1.67 (497[a]) |
| **Average Body Mass Index (BMI; kg/m$^2$)** | 23.6 (96[a]) | 25.3 (3232[a]) | 27.8 (163[a]) | 25.6 (497[a]) |
| **Average Body Surface Area (Du Bois)** | 1.72 (96[a]) | 1.78 (3232[a]) | 1.84 (163[a]) | 1.82 (497[a]) |
| **Number of women with breast cancer** | 7 (96[a]) | 685 (3127[a]) | 922 (922[a]) | N/A |
| **BIRADS classification (if available)** |  |  |  |  |
| BIRADS 0 or not available | 16 | 238[a] | – | N/A |
| BIRADS 1 | 0 | 23 | – | N/A |
| BIRADS 2 | 63 | 1878 | – | N/A |
| BIRADS 3 | 7 | 226 | – | N/A |
| BIRADS 4 | 3 | 287 | – | N/A |
| BIRADS 5 | 1 | 156 | – | N/A |
| BIRADS 6 | 6 | 424 | 922[b] | N/A |
| **MRI scanner magnetic field strength** |  |  |  |  |
| 3.0 T | 95 | 2913 | 454 | 297 |
| 1.5 T | 0 | 315 | 468 | 606 |
| 0.55 T | 1 | 4 | 0 | 0 |

[a]Number of examinations with variable(s) needed available.
[b]DUKE dataset consists of a cohort of women undergoing pre-operative breast MRI with known breast cancer. *BI-RADS* Breast Imaging Reporting And Documentation System, *T* Tesla, *MRI* Magnetic Resonance Imaging, *kg* kilogram, *m* meter, *N/A* not available

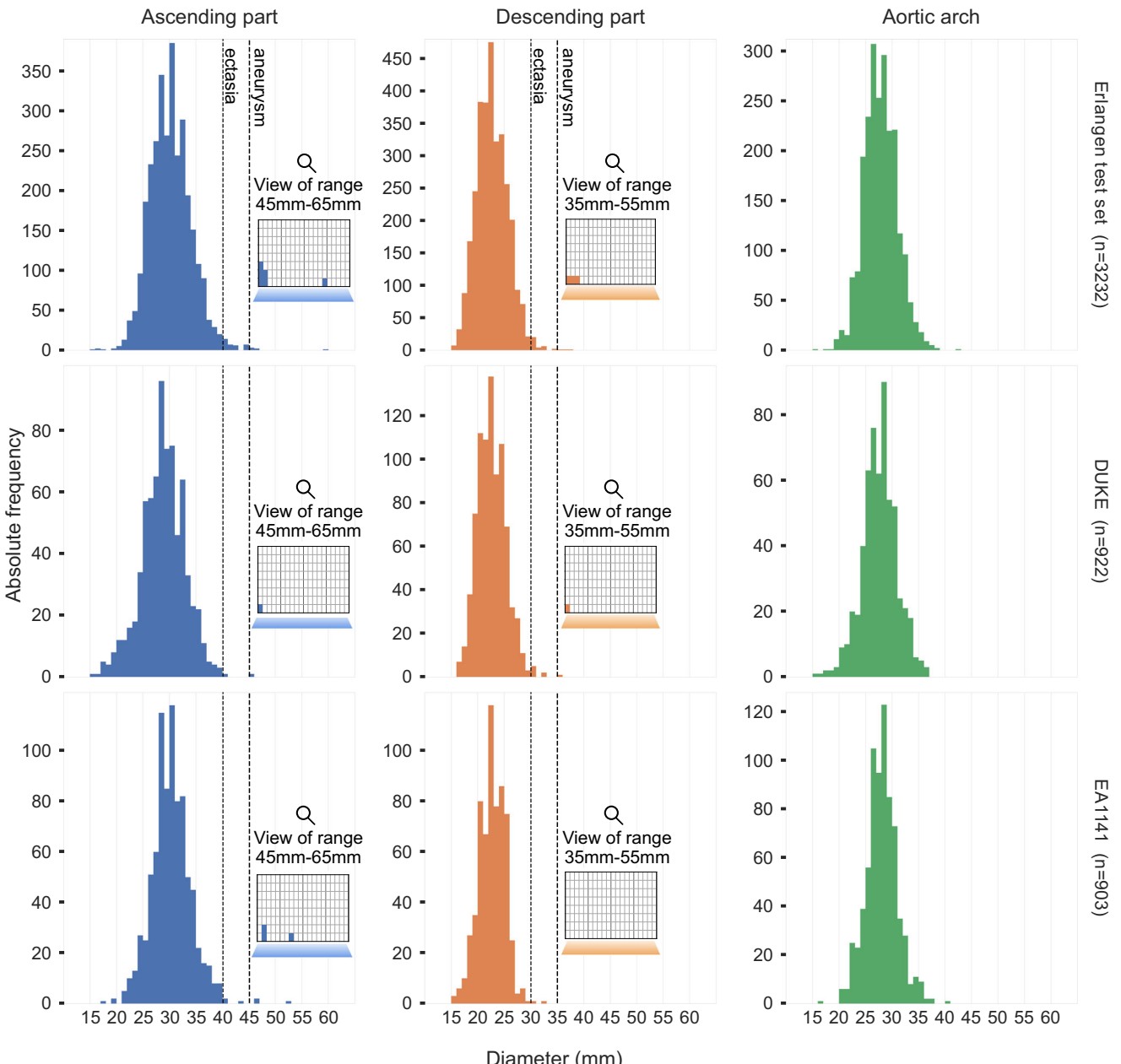

**Fig. 2 | Aortic diameter assessments by the ANN.** Histograms of the calculated thoracic aorta diameters using the proposed methodology for the three evaluation datasets ($n = 5057$ examinations in total). The dilatation (ectasia) threshold is set at 40 mm for the ascending aorta (AA) and 30 mm for the descending aorta (DA). Aneurysm threshold is set at 45 mm and 35 mm, respectively, as per the AHA2022 guidelines and values from the Framingham Heart Study. The floating subplots visualize aneurysm findings (in the 45–65 mm AA range and the 35–55 mm DA range), and each box corresponds to a single examination. Anonymized source data is provided as a source data file.

examinations (CE-CT or TTE), causing in between $246.13 and $6960 costs per detected aneurysm. Details on the analysis are given in Supplementary Table 4.

## Discussion

In this study, we show that fully-automated ANN assessments of the thoracic aorta could be a valuable supplemental screening tool for the presence of aortic aneurysms in routine breast MRI examinations that does not require additional effort. The large-scale, multi-center data analysis demonstrated both a favorable technical robustness and high clinical relevance of the proposed ANN pipeline.

The ANN approach reliably detected thoracic aortic aneurysms including a considerable number not detected during clinical routine -

with a 3.5-fold improved detection rate (increasing detected aneurysms from $n = 2$ to $n = 7$ in our cohort). The approach uses existing imaging data and, consequently, does not cause an additive burden to the patient or the healthcare system for screening. This could facilitate effortless supplemental screening for a rare yet clinically devastating disease that is otherwise not efficiently accessible, with particular relevance for women.

The ANN additionally revealed robust technical performance in three independent, multi-center, multi-vendor, and multi-field strength test datasets. The variety of test datasets highlights the practical impact, as the ANN was tested across the full spectrum of breast MRI indications (including high-risk screening, screening of women with dense breast, staging, and follow-ups).

**Table 2 | Calculated thoracic aorta diameter results on the independent test sets**

|  | Erlangen test dataset | Duke | EA1141 | Overall |
|---|---|---|---|---|
| **Breast MRI examinations (n)** | 3232 | 922 | 903 | 5057 |
| **Patients (n)** | 2258 | 922 | 480 | 3660 |
| **Aortic diameters determined (average maximum diameter, cm)[a]** | | | | |
| Ascending thoracic aorta (AA) | 2.97 (SD 0.39) | 2.84 (SD 0.41) | 2.98 (SD 0.37) | 2.95 (SD 0.39) |
| AA (women with breast cancer) | 3.04 (SD 0.37) | 2.84 (SD 0.41) | N/A | N/A |
| AA (women without breast cancer) | 2.96 (SD 0.39) | - (BC-only) | N/A | N/A |
| Descending thoracic aorta (DA) | 2.22 (SD 0.29) | 2.21 (SD 0.27) | 2.22 (SD 0.25) | 2.22 (SD 0.28) |
| DA (women with breast cancer) | 2.27 (SD 0.28) | 2.21 (SD 0.27) | N/A | N/A |
| DA (women without breast cancer) | 2.21 (SD 0.29) | - (BC-only) | N/A | N/A |

[a]Diameter is given as the maximum perpendicular diameter of the respective thoracic aortic organ section (ascending/descending). *AA* ascending thoracic aorta, *DA* descending thoracic aorta, cm centimeter, *SD* standard deviation, *N/A* data not available.

**Table 3 | Results overview of aneurysm detection on the independent test datasets**

|  | Erlangen test dataset | Duke | EA1141 | Overall |
|---|---|---|---|---|
| **Breast MRI examinations (n)** | 3232 | 922 | 903 | 5057 |
| **Patients (n)** | 2258 | 922 | 480 | 3660 |
| **Examination level detected thoracic aortic aneurysms (% of breast MRI examinations)** | | | | |
| Aneurysms (ANN predicted) | 9 (0.28%) | 2 (0.22%) | 3 (0.33%) | 14 (0.28%) |
| Ascending thoracic aorta (AA) | 6/9 | 1/2 | 3/3 | 10/14 |
| Descending thoracic aorta (DA) | 3/9 | 1/2 | 0/3 | 4/14 |
| **Recall rate (RR) / Positive predictive value (PPV)** | | | | |
| RR/PPV 1 - Ascending thoracic aorta (AA) | 1.2% / 15% | 0.22% / 50% | 0.66% / 50% | 20.8% |
| RR/PPV 2 - Ascending thoracic aorta (AA) | 0.4% / 46.2% | 0.108% / 50% | 0.44% / 75% | 55.5% |
| RR/PPV 3 - Ascending thoracic aorta (AA) | 0.187% / 100% | 0.108% / 100% | 0.33% / 100% | 100% |
| RR/PPV 1 - Descending thoracic aorta (DA) | 1.08% / 5.7% | 0.22% / 12.5% | 0.22% / 0% | 6.67% |
| RR/PPV 2 - Descending thoracic aorta (DA) | 0.155% / 40% | 0.108% / 100% | N/A | N/A |
| RR/PPV 3 - Descending thoracic aorta (DA) | 0.093% / 66.7% | 0.108% / 100% | N/A | N/A |
| Aneurysms (panel verified[a]) | 8/9 (89%) | 2/2 (100%) | 3/3 (100%) | 13/14 (92.8%) |
| Aneurysms (detected in clinical routine readings) | 2 (0.06%) | N/A | N/A | N/A |
| Negative predictive value of ANN analysis | 100% | 100% | 100% | 100% |
| **Odds ratio for thoracic aortic aneurysm in the Erlangen test dataset** | | | | |
| Odds ratio (OR) for women with breast cancer | 2.14, CI: [0.51, 8.97] | N/A | N/A | N/A |
| Odds ratio (OR) for women with breast cancer and/or breast cancer history | 2.29, CI: [0.55,9.61] | N/A | N/A | N/A |
| **Patient-level prevalence for thoracic aortic aneurysms** | | | | |
| ANN derived prevalence (panel verified[a]) | 0.31% (7/2258) | 0.22% (2/922) | 0.42% (2/480) | 0.30% (11/3660) |
| Clinical routine report prevalence | 0.09% (2/2258) | N/A | N/A | N/A |

[a]Examinations flagged as aneurysm by the ANN were considered as incorrect when a deviation of > 2 mm from the mean manual aortic diameter assessments of the radiologist panel was found and the deviation caused a re-categorization of pathological findings (i.e., dilatation vs. aneurysm) or the panel unambiguously suggested a different category for the pathological findings (i.e., dilatation vs. aneurysm or correct diameter assessment but wrong automatic selection of threshold for ascending or descending part of the thoracic aorta) independent of the deviation of diameter measurement. Amongst the Erlangen test dataset, one case was reclassified: *n* = 1 case of mis-segmentation. The negative predictive value of ANN analysis (PPV1 thresholds) was derived based on visual assessments by a radiologist panel. *AA* ascending thoracic aorta, *DA* descending thoracic aorta, cm centimeter, *CI* confidence interval, *N/A* data not available.

Our cohort revealed a higher prevalence of thoracic aortic aneurysms and larger aortic diameters in former and current breast cancer patients. However, due to the retrospective study design and the origin of the data, this finding might be biased by the selection of patients. Further, given that the confidence intervals are wide due to the low prevalence of aneurysms, no definite conclusions can be drawn regarding potential underlying causes for this finding.

Our study demonstrated that current visual reading strategies during clinical routine could benefit from the ANN screening support. This might be due to the focus of breast MRI examinations on oncologic targets, the sometimes-challenging assessment of the thoracic aorta in axial slices, and search satisfaction bias. This is further aggravated by emerging indices beyond aortic diameters (aortic height index and/or aortic surface index) with more complex calculations that must be performed separately.

The ANN is capable of running fully automatically in the background in a clinical routine workflow, providing a structured report (see Supplementary Fig. 6). This workflow integration does not require modification of any examination or assessment steps and the ANN demonstrated slightly lower measurement variation compared to two radiologists performing >1000 control measurements, suggesting potential for clinical integration. Even though organ segmentation tasks have been previously studied, including with the nnUnet utilized in our study, our approach aimed to apply such a technique for secondary data use and to organ structures that are not of focus of the examination. This out-of-the-box approach could thus be regarded as a

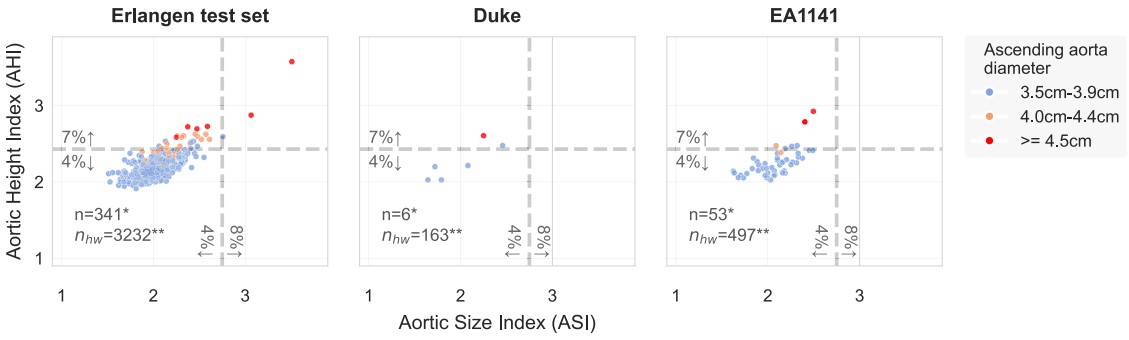

*number of patients with height/weight available and ascending aorta diameter ≥ 3.5 cm
**number of patients with height/weight available in dataset

**Fig. 3 | Ascending aortic risk analysis using advanced indexes.** Aortic height index and aortic size index for the $n = 400$ examinations with height/weight available and ascending aorta diameter ≥ 3.5 cm in the Erlangen test dataset, Duke dataset and EA1141 dataset. Anonymized source data is provided as a source data file.

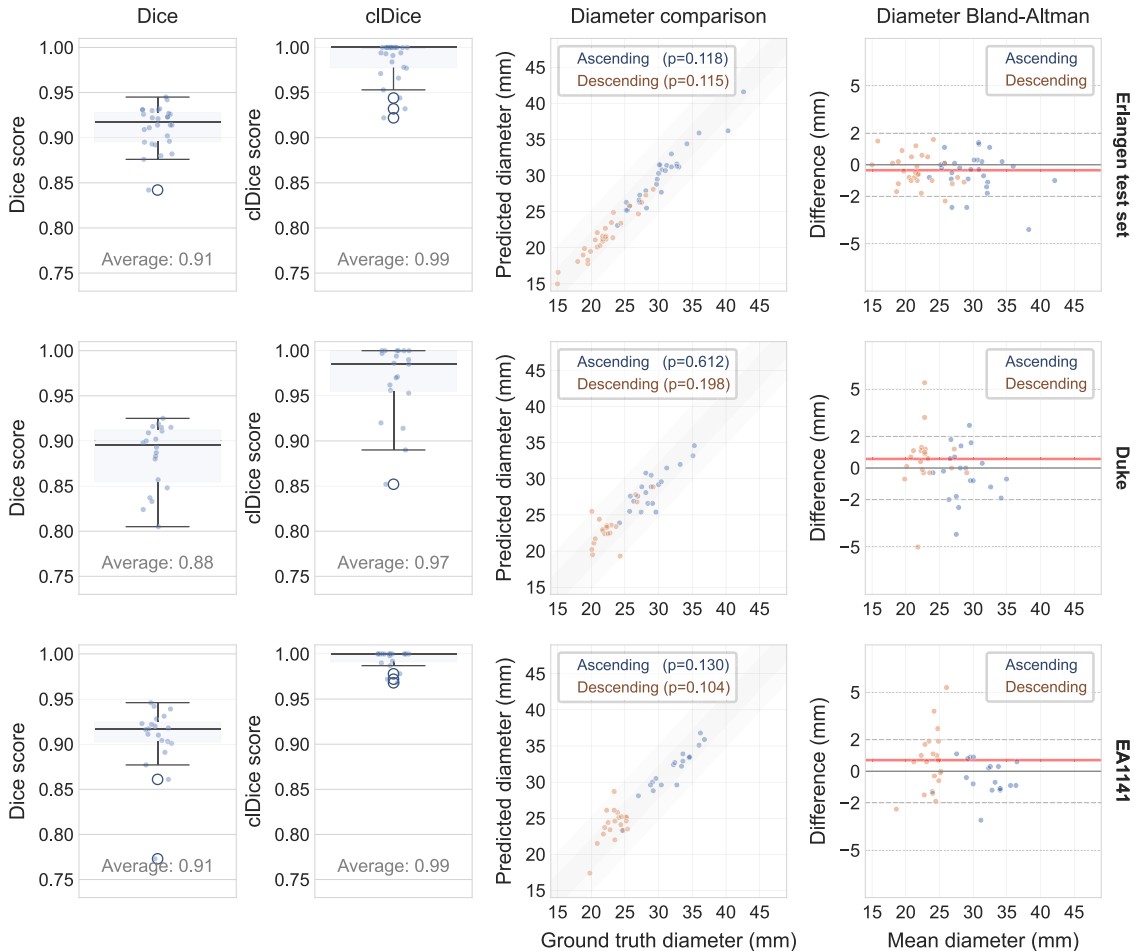

**Fig. 4 | Comparison of the segmentations produced by the ANN with ground truth annotations.** The first column plots the dice coefficient box plots (box: 25th–75th percentile, i.e., Q1-Q3; whiskers extending up to 1.5 times the interquartile range; outliers marked) for the Erlangen test set, Duke, and EA1141, demonstrate medians [Q1-Q3] of 0.92 [0.90–0.93], 0.90 [0.85–0.91], 0.92 [0.90–0.92], respectively. The second column plots the clDice box plots, which demonstrate medians [Q1–Q3] of 1.00 [0.98–1.00], 0.99 [0.96–1.00], 1.00 [0.99–1.00], respectively. The third column contains scatterplots between the AA/DA diameters calculated using the predicted and ground truth segmentations, with the fourth column being the respective Bland-Altman plots. Statistical testing was performed without correction using a two-sided t-test with paired samples and no significant difference was observed for any of the diameter predicted pairs

(Erlangen: AA $t = 1.62$, $p = 0.118$, effect size = 0.31; DA $t = 1.63$, $p = 0.115$, effect size = 0.31, DUKE: AA $t = 0.52$, $p = 0.612$, effect size = 0.12; DA $t = −1.33$, $p = 0.198$, effect size = −0.30, EA1141 $t = 1.58$, $p = 0.130$, effect size = 0.35; DA $t = −1.71$, $p = 0.104$, effect size = −0.38). There were $n = 28$ in-house ground truth segmentations used for this comparison, alongside $n = 20$ Duke and $n = 20$ EA1141 segmentations, randomly selected with the only condition that the whole aorta is visible on the image. The strongly colored shading represents the ± 2 mm range, while the lightly colored shading is the ± 5 mm range, meant to show cases where a ground truth aneurysm gets completely missed in a simulated clinical environment (as it would fall below the dilatation threshold that gets presented to the clinician in a potential clinical integration). AA Ascending Aorta, DA Descending Aorta. Anonymized source data is provided as a source data file.

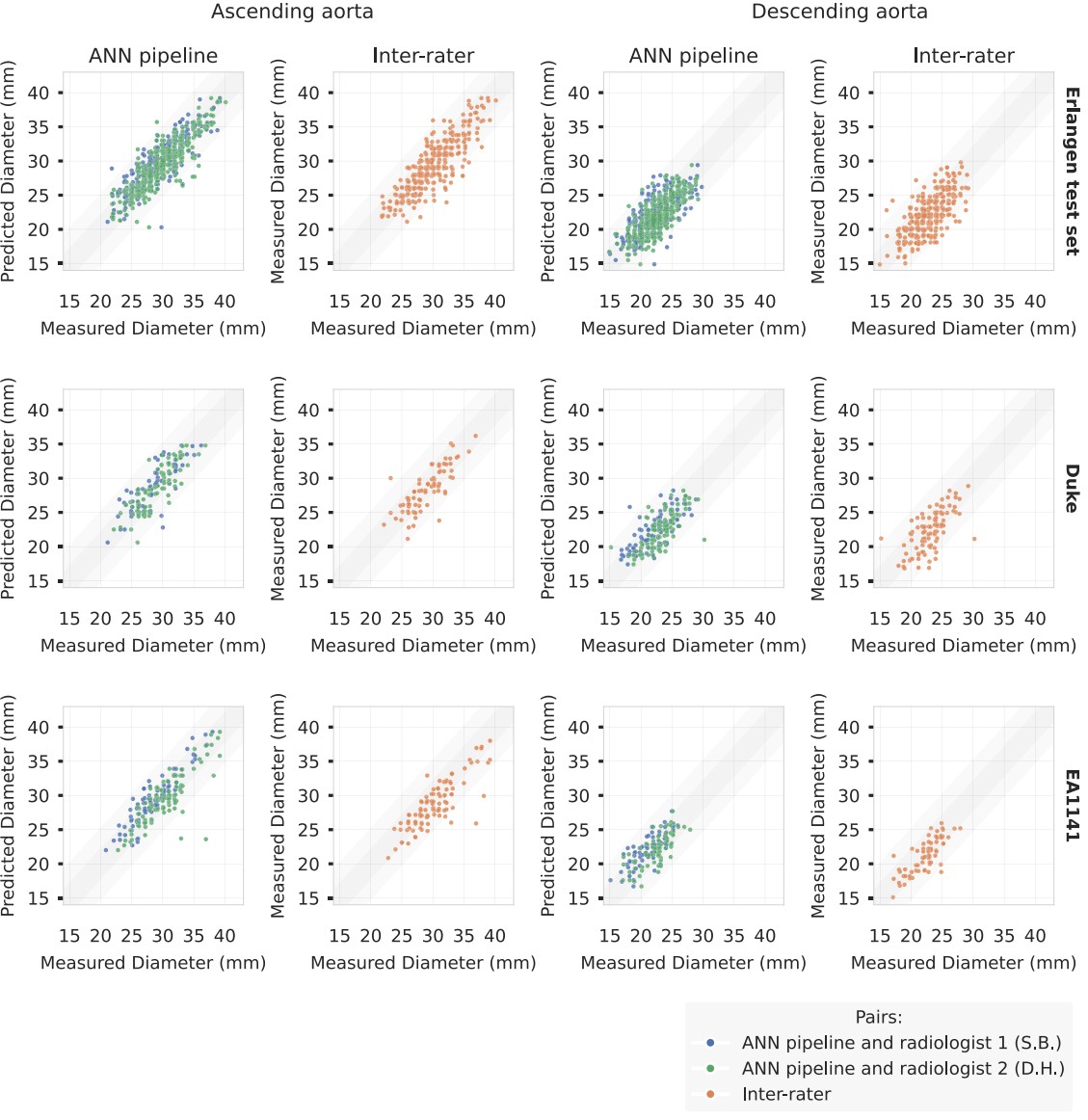

**Fig. 5 | Comparison of the diameter calculations of the ANN pipeline and manual measurements by two radiologists.** Results from a total of $n = 1060$ random control diameter measurements of the ascending or descending thoracic aorta, each performed by two radiologists. The first and third columns are scatterplots of the manual measured diameter of the radiologists and the ANN pipeline prediction for ascending and descending aorta, respectively. The second and fourth columns are scatterplots of the manually derived measurements of the two radiologists. Noise at most ± 0.2 mm was added to the radiologist measurement for this visualization only, to avoid overlapping dots from the integer measurements. The strongly colored shading represents the ± 2 mm range, while the lightly colored shading is the ± 5 mm range. Anonymized source data is provided as a source data file.

potential blueprint use case that can be applied on a broad variety of other (rare and non-rare) clinical targets. Secondary data use of large-scale imaging applications (e.g., in lung cancer screening) could strategically facilitate the detection of rare diseases that would otherwise be difficult to achieve in a cost-efficient manner. In our case, breast MRI was selected specifically because it is an established imaging modality in breast cancer imaging and is emerging as a viable supplemental screening tool in population-based breast cancer screening for women with (extremely) dense breast tissue. Breast MRI was recently suggested to be integrated into screening for women with extremely dense breast tissue by the European Society of Breast Imaging (EUSOBI)[16].

The ANN achieved a high Dice coefficient (0.88–0.91) over different sites, vendors, all currently established clinical MRI field strengths (0.55-3 T), and different acquisition protocols, as well as contrast agents used during the examination. It exceeded previously reported Dice coefficients for aortic assessments in non-dedicated thoracic MRI, e.g., without ECG-gating (0.85)[20], while studies using dedicated ECG-gated MRI reported similar or even higher Dice coefficients, yet in smaller sample sizes (0.91–0.95)[21–23]. It also achieved a comparably, slightly lower deviation from manual measurements (− 0.52 mm vs. 0.9 mm)[20].

Data on the prevalence of thoracic aortic aneurysms is limited. Girotti et al. reported an incidence of thoracic aortic dilatation in low dose computed tomography (LDCT) lung cancer screening of 4% and thoracic aortic aneurysms of 0.25%[24], which is in the range of 0.16–0.34% reported for thoracic aortic aneurysms by Kuzmik et al.[2]. The ANN derived prevalence of thoracic aortic aneurysms in our patient cohort (0.31/0.22/0.42%) aligns with existing literature. However, to our knowledge, our study is the largest to report specific prevalence data for women based on volumetric aortic assessments.

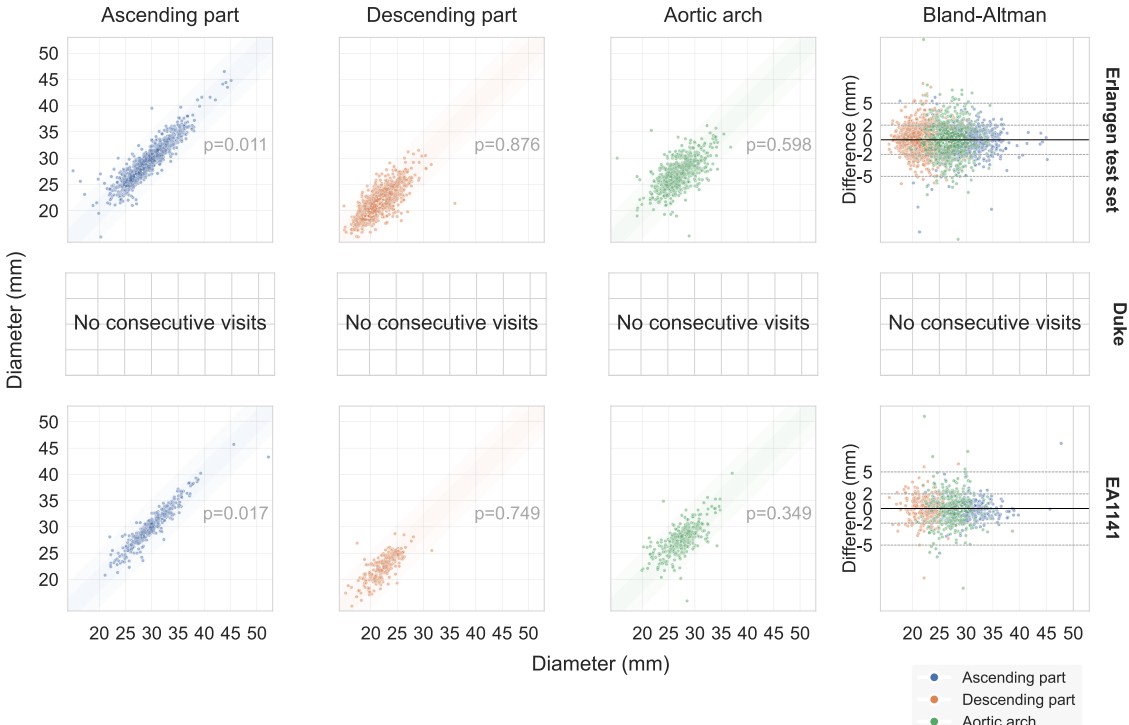

**Fig. 6 | Re-test reproducibility of aortic segmentations.** Calculated diameter agreement between two consecutive examinations of the same patient, for all possible consecutive pairs. There were $n = 974$ consecutive examination pairs in the Erlangen test dataset (belonging to $n = 585$ patients), $n = 0$ for Duke, and $n = 423$ pairs in EA1141 (belonging to $n = 419$ patients), which resulted in 3382 AA/DA/arch diameter pairs, after excluding those where the calculation was not possible. The expectation is that there are typically no extreme variations in the diameter calculation between visits. The strongly colored shading represents the $\pm 2$ mm range, while the lightly colored shading is the $\pm 5$ mm range, meant to show cases where a ground truth aneurysm gets completely missed in a simulated clinical environment (as it would fall below the ectasia threshold that gets presented to the clinician in a potential clinical integration). Statistical testing was performed without correction using a two-sided t-test with paired samples (Erlangen: AA $t = -2.55$, $p = 0.011$, effect size $= -0.08$; DA $t = 0.16$, $p = 0.876$, effect size $= 0.01$; arch $t = -0.53$, $p = 0.598$, effect size $= -0.02$, EA1141: AA $t = -2.39$, $p = 0.017$, effect size $= -0.12$; DA $t = -0.32$, $p = 0.749$, effect size $= -0.02$; arch $t = -0.94$, $p = 0.349$, effect size $= -0.05$). The last column presents the respective Bland-Altman plots. AA Ascending Aorta, DA Descending Aorta. Anonymized source data is provided as a source data file.

Uncovering the overrepresentation of women with breast cancer among cases with thoracic aortic aneurysms is of potential high clinical relevance. In the literature, patients with lung cancer have been shown to have a higher rate of abdominal aortic aneurysms[25], as have prostate cancer patients[26]. Moreover, some studies suggest that certain oncologic therapies may potentially contribute to aneurysm growth rates, though conflicting reports exist[27]. This highlights the clinical relevance of our approach in addressing a relevant clinical target group with an effortless support tool for clinicians. While the absolute diameter differences were too small to derive direct clinical implications, a recent study on lung cancer screening participants reported aortic assessments to be the second most prognostically relevant factor for predicting 10-year mortality[28].

The ANN measurements demonstrated promising consistency across different timepoints after GBCA injection as well as promising longitudinal consistency of ANN across repeat examinations. However, variations were also present. Few studies have investigated the natural course of thoracic aortic aneurysms and dilatation, with the recommendations on monitoring being an ongoing discussion[5,6]. In this regard, despite being tempting, using the ANN for monitoring disease progression might be limited due to the low growth rates (0.1–0.3 mm/year[29]) and the limited image resolution of MRI.

The present study has the limitations of only including retrospective data; thus, we cannot draw direct conclusions about the prospective clinical application. However, the Erlangen test set comprised a consecutive 5-year cohort of patients, and we enriched the test data with multi-vendor public datasets. Further, it is possible that in a prospective clinical study (in which the radiologists would be aware that in parallel to them an AI is searching for aortic diseases) a diagnostic awareness bias might potentially be introduced, limiting real world insights, compared to our evaluation in which the radiologist's reports from clinical routine were used as real-world indicators of reporting frequency of aortic diseases.

Our study has several other limitations. Underreporting data was based solely on the Erlangen independent test, as clinical reports from the external test datasets are not publicly available. While we aimed to reflect the variety of acquisition settings, vendors, and field-strengths in the independent test sets, some MRI machines/settings/protocols might be insufficiently represented, and, consequently, generalizability across all potential settings cannot be fully assessed. In some cases, parts of the aorta, e.g., upper arch, were not fully covered/depicted by the chosen field of view (FOV) or due to the image quality of the breast MR examination, which may restrict the method's applicability, depending on acquisition settings and imaging hardware. Moreover, more sophisticated approaches might allow for further refinement of the ANN, however the chosen nnU-Net approach has demonstrated superior performance in several open challenges[30]. No testing was performed on artificially noisy or perturbed images, the training data was limited, and no explainability methods[31] were used, which could shed light on which areas the model considers important for producing the segmentations. Further, the dual-use approach of the study meant that the MRI examinations were not optimized toward aortic vessel assessment. We, therefore, included a variation threshold of 2 mm in order to account for potential overestimations as previously suggested[32]. This limitation is important as abnormalities (either dilation or aneurysms) close to the thresholds could otherwise

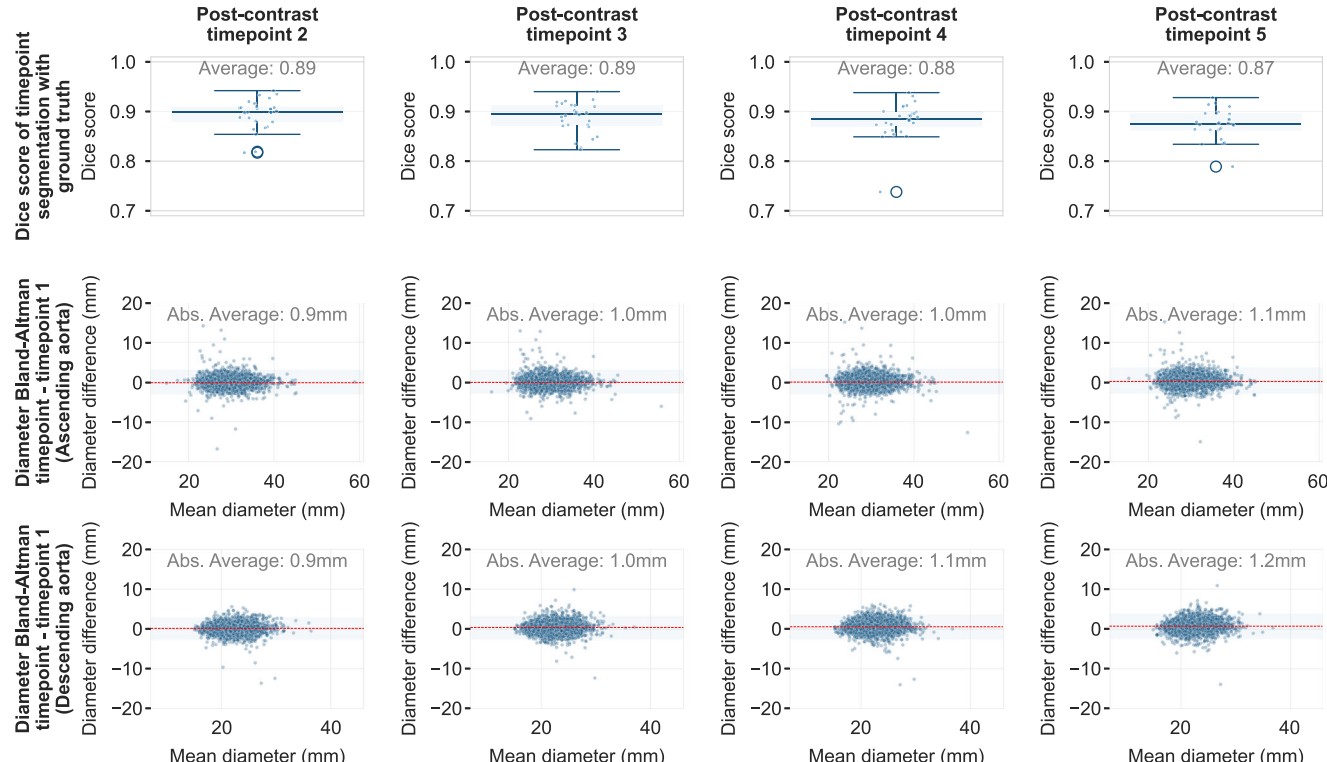

**Fig. 7 | Reproducibility of aortic segmentations on breast MRI for different timepoints after GBCA-administration in the Erlangen test set.** DICE coefficient reproducibility over the different timepoints after GBCA administration (T1-weighted sequence) for the $n = 28$ cases with in-house ground truth segmentations. Box plots (box: 25th–75th percentile, i.e., Q1–Q3; whiskers extending up to 1.5 times the interquartile range; outliers marked) for timepoints 2, 3, 4, and 5, demonstrate stable medians [Q1-Q3] of 0.90 [0.88–0.91], 0.89 [0.87–0.91], 0.89 [0.87–0.90], 0.88 [0.86–0.90], respectively. The average variation of the diameter (mm) (as measured for the full $n = 3232$ Erlangen test set) for ascending and descending thoracic aorta was in the range of 0.9–1.1 mm and 0.9–1.2 mm (exact values are present directly on the figure), respectively. The colored area in the Bland-Altman plots represents the 95% prediction interval. Anonymized source data is provided as a source data file.

potentially go unnoticed due to the pulsation effects close to the heart. Given the encouraging results, future work might investigate as well imaging techniques that overcome the non-ECG imaging limitations by adapting cardiac MRI techniques for breast MRI. This might be achieved by adding ultrafast cine SSFP sequences to the breast MRI protocol[33], which could allow for self-gating without ECG leads. This could be beneficial as ECG-leads are associated with some challenges such as preparation time, costs, inaccuracies associated to artifacts caused by the MRI hardware[34,35] and could be especially cumbersome in prone-positioned breast MRI examinations.

Depending on the thresholds chosen for recalls, some patients will undergo additive examinations due to a false positive call of the ANN. This might cause additive burden and costs, e.g., due to additive ultrasound or computed tomography examinations. Our analysis indicated additive costs in the range of $246.13 to $6,960 per detected aneurysm, which is, however, considerably lower when compared to e.g., the costs per detected cancer in high-risk women invited to undergo breast MRI (with approximately $35,085 [£28,284] per detected cancer[36]). These costs might be considered from a health economics perspective when selecting thresholds in the future, in order to balancing re-calls and false positives for the ANN, which goes beyond the scope of this manuscript. No long-term follow-ups of the patients with thoracic AA and AD were available, and we cannot assess the clinical course of the disease in those patients. Lastly, we focused only on using the most robust and established routine acquisition technique in breast MRI[18,37].

In conclusion our study demonstrated fully-automatic background screening for aortic aneurysms using breast MRI examinations, indicating hidden potential for dual use of the acquisitions.

## Methods

### Study design and participants

The study was approved by the institutional review board (IRB) of the Friedrich-Alexander-University Erlangen-Nurnberg, Germany, waiving the need for informed consent, as working with deidentified retrospective data from clinical routine examinations only. The multicentric, retrospective study included a total of $n = 5153$ datasets as follows: $n = 3328$ breast MRI examinations originated from patients examined at the Erlangen University Hospital (Erlangen, Germany) and additive data originated from two independent large-scale trials: $n = 922$ breast MRI studies were included from the "Duke-Breast-Cancer-MRI" data set (referred to as "DUKE" in this study)[38] and $n = 903$ breast MRI studies from the multicentric EA1141 trial[39], with the latter trial reporting to having included over $n = 50$ independent study-sites.

### Training dataset

A training dataset was derived from a non-consecutive cohort of $n = 96$ adult patients (aged > 18 years) with clinical indication for breast MRI examined at the Erlangen University Hospital (Erlangen training dataset). Data were acquired between 2015 and 2020. The Erlangen training dataset was ensembled in a manner that represents variations in breast MRI examinations and different enhancement patterns of the aorta. Specifically, cases with and without full coverage of the aortic arch, different ages and MRI scanner generations were included in the training dataset.

### Independent testing datasets

For independent testing of the ANN after the final development, three independent datasets were established as the following: A consecutive

independent cohort (excluding $n = 217$ examinations belonging to the ensembled $n = 96$ patients whose examinations were included in the training dataset) of $n = 3232$ examinations ($n = 2258$ adult patients) undergoing breast MRI from 2015–2020 at the Erlangen University Hospital – the Erlangen test dataset. Two additional multisite independent testing datasets comprising of the aforementioned DUKE and EA1141 datasets were evaluated as provided by the cancer imaging archive[40], including additional $n = 1825$ independent breast MRI examinations and comprising all major MRI manufacturers (Philips, Best, The Netherlands; Siemens Healthineers, Erlangen Germany; GE-Healthcare, USA) and MRI devices from different clinically routine field strengths (1.5 T and 3 T) and scanner generations, as described in Table 1 and Supplementary Table 1.

Herein, the datasets reflect the diverse settings breast MRI is performed in. The Erlangen test dataset includes the clinical routine variety of current indications for breast MRI (high-risk screening, (pre-operative) staging, follow-up, surveillance) over a longitudinal cohort spanning 5 years. The DUKE dataset includes breast cancer patients with known disease only, while the EA1141 dataset includes a cohort of women with dense breasts undergoing breast MRI screening.

All breast MRI examinations were conducted on routine MRI scanners, commonly including a multiparametric breast MRI protocol which comprises dynamic contrast enhanced (DCE) T1-weighted (T1w) acquisitions before and after intravenous administration of GBCA, with varying additive sequences (namely, T2-weighted and/or diffusion weighted imaging [DWI]) being acquired in the different institutions. Details of the acquisition protocols for the Erlangen dataset are provided in the supplemental material, while the respective protocols for the multi-site trials can be accessed through the Cancer Imaging Archive and/or provided at the given reference (DUKE data set[38] and EA1141 trial[39]). No data was excluded from the analysis based on an assessment of image quality of organ coverage, although $n = 50$ examinations were excluded from the EA1141 ($n = 22$ examinations for technical issues and $n = 28$ because they were acquired with an ultra-fast approach outside the scope of our study).

### Data preparation
From the varying breast MRI datasets, the T1w-contrast-enhanced sequences reflecting an early phase (until approximately ~120 sec. after GBCA-injection, as far as derivable for the external datasets) were used for further processing. This approach was selected in order to max-imize the applicability of this study across acquisition protocols, especially giving the increasing role of abbreviated breast MRI approaches, which can only include a single timepoint (commonly up to two minutes) after contrast agent application. The T1w contrast-enhanced sequence is the core of breast MRI and reflects a broad applicability of the thoracic aortic disease screening approach, ranging from abbreviated breast MRI protocols currently assessed for breast screening[18,41] to full multiparametric diagnostic breast MRI protocols[17]. For further assessment of the robustness of the ANN towards the variance of acquisition timepoints after contrast agent injection, the Erlangen test data was further explored. The dynamic contrast-enhanced (DCE) sequence in the breast MRI protocol is composed of five individual T1-weighted acquisitions after GBCA-injection, which were individually provided to the ANN pipeline.

### Preprocessing
The study workflow is depicted in Fig. 1. The only training pre-processing step performed was minimal and consisted only of resampling early (commonly 1st timepoint after injection of GBCA, in some trials the abbreviated protocol was not further specified) post-contrast DCE-MRI acquisition to a NIfTI image with spacing (0.85 mm, 0.85 mm, 1.5 mm). The image array was also flipped if necessary to the same orientation as the training data, i.e., ensuring that the origin direction vector is always positive and the

corresponding patient position is the same, using only image meta-information (origin and direction).

### Segmentation of the thoracic aorta
The preparation of the image data included a volumetric aortic seg-mentation for both the training and the independent test data sets using the aforementioned GBCA-enhanced T1-weighted sequences during the arterial contrast phase by experienced trained research personnel (medical student, JG, providing segmentations in breast MRI for one year) under the supervision of a board-certified radiologist with a focus on oncologic imaging and breast MRI and 10 years of experience (SB).

The segmentation was performed using the axial slices and additional planes reconstructed by the annotation software tool (Slicer 5.03[42]). The following steps and guidelines were used: Using the draw function of the software tool, segmentation of the aorta was per-formed adopting the recommendations given for aortic assessment in imaging, resulting in the inner edge-to-inner edge (I-I) technique[43,44] to be leading as suggested to most closely match the routine leading edge-to-leading edge (L-L) method from echocardiography. The con-tours of the thoracic aorta were outlined in a slice-by-slice manner following from the aortic root (commonly the bulbus/sinotubular junction, depending on image acquisition) to the lowest visible part of the descending thoracic aorta (commonly at the diaphragm). In some cases, parts of the aorta, e.g., the upper arch, were not fully covered/depicted by the chosen field of view (FOV), or depending on the image quality of the breast MR examination.

### Training of the ANN
The architecture of the ANN was a Convolutional Neural Network (CNN), based on the award-winning nnU-Net segmentation framework[30]. nnU-Net works by "fingerprinting" the training dataset properties, being able to self-configure itself for preprocessing, train-ing, and postprocessing based on the corresponding task. A combined loss was utilized, as the summation of dice and binary cross-entropy losses[30]. The underlying network architecture is a U-Net architecture with plain convolutions for feature extraction, trained using stochastic gradient descent. We utilized the high image resolution 3D config-uration (with rules derived for non-CT anisotropic data with median image size $447 \times 445 \times 120$ and spacing $0.85 \times 0.85 \times 1.5$ mm), with z-score normalization, 1000 epochs, 250 iterations per epoch, 50 validation iterations per epoch, initial learning rate 0.01, weight decay $3*10^{-5}$, batch size 2, and patch size $56 \times 192 \times 224$ (determined by nnU-Net's self-configuring process). We additionally included the following augmentations to aid adaptation over a wide range of MR data from different MRI scanners and institutions: (i) rician noise augmentation (probability 30%) which can potentially mimic the noise found in MRI data more appropriately than gaussian noise[45], (ii) random MRI spike artifacts (probability 10%) which can be caused by some mal-functioning MRI machines resulting in sharp disruptions in the image, (iii) random MRI bias field artifacts (probability 10%) which can mimic variations in image intensity caused by non-uniformities in the mag-netic field, resulting in uneven brightness or contrast, (iv) random MRI ghosting artifacts (probability 10%) which are false image artifacts alongside the true image occurring due to motion or faulty scanners, and (v) random MRI motion artifacts (probability 10%) which can manifest as blurring in the image due to patient movement[46]. Images were resampled using third-order spline and segmentations with linear interpolation, utilizing nearest neighbor for out-of-plane areas. The model was trained on an NVIDIA V100 32GB on a system with 32 cores and 256GB of RAM.

### Calculating the aortic diameter
**Additional preprocessing.** As a first step to the calculation, the pro-vided segmentation is resampled to isotropic spacing (using its

original minimum spacing in all dimensions, i.e., 0.85 mm on the pre-process images). Each axial slice is processed if needed to keep only the two largest segmentation components, which are most likely to belong to the aorta. Binary closing and hole-filling operations are also performed on the segmentation.

**Centerline determination.** A mirroring operation is performed on the segmentation in the z-direction, which helps extend the resultant centerline to the edge. The centerline is calculated with the skeletonization operation[47], and centerline points that have overlap with the original segmentation (excluding the mirrored parts) are kept. The Dijkstra algorithm is applied to ensure that a single continuous line is kept[48]. Finally, to reduce processing overhead, we remove centerline points that are within 2 mm of other points. This process is repeated for each unconnected segment of the segmentation. For small segmentation parts of height < 25 mm, typically resulting from the aortic arch being outside the field-of-view and the ascending aorta being a separate segment, a simpler process is followed instead, where the segmentation is eroded only in the x and y dimensions, until a centerline remains.

**Plane selection.** In order to define a plane for each centerline point, a normal vector needs to be calculated. For each point, we calculate the normal vectors defined by the current and the previous point, alongside the one defined by the current and the next point, and average them at each dimension (point order is dictated by the Dijkstra algorithm). This process is repeated for the 3 previous and 3 next centerline points (if they are within 1 cm), and the final normal is averaged across all calculated normal vectors. For each defined plane, 27 different small variations are tested (by shifting the normal vectors by [− 15%, 0, + 15%] at each direction), and the combination that leads to a segmentation with the highest circularity is kept. If no combination resulted in an acceptable circularity, then the plane is discarded (circularity threshold: 85% or 75% if the plane normal in the z-direction is ≤ 0.5, to account for the less circular shapes in the z-direction due to the larger slice thickness).

**Plane categorization.** To categorize planes into thoracic AA/DA/arch, we first attempt to categorize points based on other points. If a centerline point has other points more than 3 cm to its right in the y-direction, then it is part of the AA, and the reverse for DA. Similarly, if it has points more than 12 cm above it in the z-direction, then it is part of the DA. Then in a second round, points that are close to already categorized points, or roughly the same z-axis line, or in-between categorized points, with similar normal vectors, inherit their neighbor's category. In a third round, the normal vectors are checked, and if the (normalized) normal vector is < 0.4 in the z-direction then it is part of the aortic arch, while a y-direction value of >0.3 indicates AA and < − 0.3 indicates DA. Finally, an additional round of inheriting a similar neighbor's category is performed.

Plane categorization appropriateness was assessed by selecting a subset of n = 50 cases in which the anatomic landmarks (supra-aortic branches) were visible. Manual assessment of the mean distance in between the anatomic landmarks and the automatic plane categorization were conducted as follows. For the beginning of the aortic arch (adjacent to the ascending aorta), the start of the brachiocephalic artery branch was marked in the open-source software 3D Slicer (Version 5.6.2) by choosing the point on the upper part of the aorta where the brachiocephalic artery starts leaving the aorta. From this point, the nearest centerline point of the aorta was selected. The distance between the selected centerline point and the first centerline point categorized as aortic arch (adjacent to ascending aorta) was calculated. Analogously, for the end of the aortic arch (adjacent to the descending aorta), the end of the left subclavian artery branch was

marked, and the distance between the respective centerline point and the last aortic arch centerline point was calculated.

**Diameter calculation.** With a known category, point, and normal vector, the segmentation array can be rotated accordingly, and the diameter on the new defined slice can be calculated. In order to account for the non-ECG triggered nature of breast MRI examinations the diameters calculated were considered with a 2 mm threshold of variance, as previously reported.

## Definition of cut-off values for determining presence of aortic disease (dilatation and aneurysms)
Using the determined aortic diameter we quantitatively assessed the presence of thoracic aortic dilatation and thoracic aortic aneurysm separately for the ascending and descending thoracic aorta. Definitions of enlarged diameters of the thoracic aorta were selected according to international guidelines and literature as the following.

**Ascending thoracic aorta (AA).** Given the sharp increase in risk of dissection relative to a control aortic diameter of ≤3.4 cm for (i) a diameter of 4.0 cm to 4.4 cm, associated with an 89-fold increased risk, and (ii) a diameter of ≥4.5 cm, associated with a 6000-fold increased risk, as per the AHA 2022 guidelines women-specific cut-off values[5], we used these thresholds accordingly in our study: thoracic ascending aortic dilation (AA dilatation) at 4.0–4.4 cm[5] and thoracic ascending aortic aneurysm (AA aneurysm) at an aortic diameter ≥4.5 cm[5].

**Descending thoracic aorta (DA).** For the descending thoracic aorta (DA) presence of dilatation was defined as 3.0–3.4 cm and thoracic descending aortic (DA) aneurysm was defined as ≥3.5 cm following the data from the Framingham Heart Study[49] with an average diameter of $23.1 \pm 2.6$ mm reported for the diameter of women and with dilatation being considered at diameters adding to 2 standard deviations to the mean diameter and aneurysm being defined at 150% of the vessel in concordance with the American college of radiology (ACR) Incidental Finding Committee[50].

## Risk analysis
Risk assessment of the thoracic ascending aortic diseases was performed using the aortic height index (AHI) and the Aortic Surface Index (ASI), as per the more recently suggested individual adjusted nomogram data for the ascending thoracic aorta, which were calculated as in literature[51].

Aortic Surface Index (ASI; cm/m²)[51]:

$$\frac{Aortic\ Diameter\ (cm)}{Body\ Surface\ Area\ (m^2)} \tag{1}$$

Body Surface Area (BSA) herein was calculated using the Du Bois formula[52].

Aortic Height Index (AHI; cm/m)[51]:

$$\frac{Aortic\ Diameter\ (cm)}{Height\ (m)} \tag{2}$$

We utilized the same thresholds, denoted annual risk of complications, as reported in literature[51]. Specifically, AHI > 2.43 marks an elevation in risk from 4% to 7%, while ASI > 2.75 marks an elevation in risk from 4% to 8%[51].

## Evaluation of the ANN
Evaluation of the ANN took place using n = 5057 multi-vendor breast MRI examinations for independent testing, corresponding to n = 3660 patients, from 3 independent sources spanning trials that involved n > 50 sites. In order to evaluate the utility and applicability of the

system, different experiments were performed covering (i) the ability to fully automatically segment and assess aortic disease (dilatation and aneurysms) in breast MRI using an ANN, (ii) comparison to prevalence data and to clinical reporting frequency, and (iii) investigating potential effects of breast cancer on thoracic aortic disease.

**Technical assessments of the ANN using volumetric segmentations.** For evaluation of segmentation performance $n = 68$ manual volumetric segmentations were produced in the independent test datasets ($n = 28$ for the Erlangen test dataset, $n = 20$ for DUKE, $n = 20$ for EA1141) belonging to random examinations, using the same process described before. Segmentation performance was evaluated using the Dice coefficient, which is the gold standard for segmentation performance assessment[30,53,54]. Further, centerline Dice (clDice) evaluation is performed, due to its applicability in problems with tubular structures[54,55], in order to gauge the ability of the segmentation to annotate large parts of the aorta and follow its centerline, instead of being performant at a part of it and worse in others. As an additional indirect measure of performance, consecutive examinations belonging to a single patient were identified, and each pair of diameter assessments were plotted. The assumption is that despite variations in aorta diameter being possible between different examinations of the same patient (due to natural changes that come with age, environmental and lifestyle factors, or aortic disease), the calculated diameter shouldn't typically see extreme variations between visits. According to the literature, the expected rate of change due to age alone is not drastic, typically reported as <2 mm/decade[56,57], which should also accommodate for changes due to systole/diastole. As such, the rate of repeat measurements that fall within 2 mm of each other is also used as an indirect measure of performance and stability. Significance testing was performed between the two sets of diameters at different time points.

**Evaluation across different timepoints after GBCA-administration.** The ANN pipeline performance was evaluated in the Erlangen test datasets on all different timepoints after GBCA-administration (timepoint 1–4) to assess in influence of timing on the DICE coefficients and the determined diameters of the thoracic aorta.

**Visual assessments using a panel review.** All examinations of the three independent test datasets were reviewed case-by-case by the expert panel consisting of two board-certified radiologists (SB with >10 years of experience in oncologic imaging and breast MRI and DH with >15 years of experience in breast MRI and general radiology). The datasets were shown to the reader, and they were required to assess a) the anatomical appropriateness of the ANN-derived segmentation and b) the appropriateness of the plane selection/categorization for the diameter determination. The reading was done independently by both readers, followed by a subsequent consensus reading for all cases in which the two readers provided diverging ratings. In case of the ANN providing erroneous results (e.g., segmenting the wrong organ or selecting the ascending aorta threshold in the descending aorta), the type of error was noted for further descriptive analysis of the frequency of error types.

**Evaluation using manual diameter measurements.** All cases for which ANN indicated at least reaching dilatation or aneurysm thresholds for the ascending or descending aorta ($n = 93$), a manual measurement of the aortic diameters was performed independently by the expert panel consisting of the same two board certified radiologists (SB with >10 years of experience in oncologic imaging and breast MRI and DH with >15 years of experience in breast MRI and general radiology).

Further, the panel was asked to individually and independently determine the diameter in a randomly chosen subset of $n = 1060$

ascending or descending aortas in $n = 560$ patients for control purposes. The manual diameters were registered and compared to the ANN-derived diameters. The imaging data was loaded into the 3D Slicer software (3D Slicer, Version 5.03 32) and by using the diameter assessment function of the open-source tool the diameters were manually measured using the inner edge-to-inner edge (I-I) approach[43,44].

**Prevalence assessment.** The ANN provided measurements and assessments were compared to prevalence as reported in the literature. For the Erlangen test set, the prevalence of thoracic aortic aneurysms and dilatation was additively compared to the clinical reporting frequency. The data was statistically analyzed by comparing the prevalence calculated by the true positive thoracic aortic aneurysms and dilatation cases provided by the ANN to the thoracic aortic aneurysms and dilatation cases indicated by clinical routine reporting. The data was statistically evaluated to assess the effectiveness of the ANN by calculating the decreased or increased detection rate of thoracic aortic aneurysms in comparison to the current clinical real-world reference standard defined as the human routine reading[58]. In order to evaluate over- or underreporting of thoracic aortic aneurysms and dilatations in breast MRI by the ANN, the Erlangen test dataset was further explored with regard to the following tasks to determine prevalence during clinical routine:

- Clinical radiologists' reports were crawled with regards to the words "aneurysm" (OR) "aorta" (OR) "ectasia" (OR) "extension" (OR) "dilatation" (OR) "aneurysmatic" (OR) "dilated" (OR) "widened" in order to determine under- or over-reporting of aortic disease as compared to the automated assessments. Search was performed in German, reflecting the language used at the hospital where the Erlangen test dataset originated.
- All cases with thoracic aortic aneurysms described in radiologists' clinical reports were additively re-evaluated by two radiologists.

Lastly, for the Erlangen test set, which includes both patients with breast cancer and patients with benign findings, potential variations of aortic diameter between these two patient cohorts were examined. Significance testing was performed for the two sets of measurements and their distributions. The odds ratio of discovering an aneurysm between the malignant and benign cohorts is also reported, utilizing only aneurysms verified by the radiologist panel.

**Risk assessment and threshold variations.** For the examinations with height/weight available, the ones with predicted AA diameter of ≥35 mm were used for risk evaluation, as the literature defines AHI/ASI for AA diameters above this threshold[51]. The number of examinations with elevated risk were identified, and cases with high risk but not an aneurysm classification were examined by the radiologist panel.

Additively different positive predictive value (PPV) thresholds were evaluated: PPV 1 was assessed for presenting all findings exceeding dilatation thresholds to the radiologists for review (ascending aorta = 40 mm, descending aorta = 30 mm)[5]; PPV 2 was assessed presenting all findings that exceeded AHA guideline aneurysm thresholds with an additive two millimeter security range (43 mm for the ascending aorta, 33 mm for the descending aorta) and PPV 3 was assessed presenting those cases exceeding the AHA aneurysm definitions[5] only.

Using the different PPV thresholds, we also calculated how many additive imaging examinations and costs are needed for clarification of findings indicated by the application of the ANN pipeline. This was performed considering either the use of contrast enhanced computed tomography (CT) (assuming $246.13 according to ref. [59]) or transthoracic ultrasound (assuming costs of $464 according to ref. [60]) in the clarification process.

**Statistics.** Confidence intervals used in this study pertain to 95% confidence intervals. Statistical tests between sets of calculated diameters from the same patients were performed using a t-test with paired samples[61]. Tests between sets of calculated diameters from different patients were performed using the Mann–Whitney U test[62]. The Kolmogorov–Smirnov test was used to compare distributions[63]. Significance was determined at $p < 0.05$.

## Reporting summary

Further information on research design is available in the Nature Portfolio Reporting Summary linked to this article.

## Data availability

The Erlangen breast MRI datasets originate from clinical routine, protected by national privacy laws and hence are not publicly available or shareable with third parties due to the retrospective nature of the study performed and the associated IRB approval for the study limiting the data use. The additive external and publicly available datasets (at the time of the study) used in the study are the following, with the detailed regulatory and licensing information to be found under the respective source. The external datasets and their (future) availability and conditions of availability are not under the control of the authors of this study. At the time the study was conducted, the accessibility was as follows: The EA1141 breast MRI datasets were accessible (at the time of the study) via The Cancer Imaging Archive[39]. The dataset was accessed via the Cancer Imaging Archive homepage. The data was accessible at the time of the study under the CC BY 4.0 license. The Duke-Breast-Cancer-MRI breast MRI datasets were accessible (at the time of the study) via The Cancer Imaging Archive[38]. The dataset was accessed via the Cancer Imaging Archive homepage. The data was accessible at the time of the study under the CC BY-NC 4.0 license. Source data of the figures are provided. Source data are provided in this paper.

## Code availability

The code can be accessed under the "CC BY-NC 4.0" license[64] as the following: Bounias, D. MIC-DKFZ/aorta-aneurysm-assessment: 2025-04-08 (2025-04-08), https://doi.org/10.5281/zenodo.15180322. (Zenodo, 2025).

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

## Author contributions

D.B.: Conceptualization, Data Curation, Formal Analysis, Investigation, Methodology, Software, Validation, Visualization, Writing – Original Draft Preparation, Writing – Review & Editing; T.F.: Conceptualization, Data Curation, Methodology, Software, Validation, Writing – Original Draft Preparation, Writing – Review & Editing; L.B.: Data Curation, Writing – Review & Editing; J.G.: Data Curation, Writing – Review & Editing; L.A.K.: Data Curation, Writing – Review & Editing; A.L.: Data Curation, Writing – Review & Editing; H.S.: Data Curation, Writing – Review & Editing; J.E.: Data Curation, Writing – Review & Editing; D.H.: Data Curation, Writing – Review & Editing; D.S.: Data Curation, Writing – Review & Editing; R.F.: Methodology, Writing – Review & Editing; P.N.: Methodology, Writing – Review & Editing; B.K.: Methodology, Writing – Review & Editing; E.W.: Data Curation, Supervision, Writing – Review & Editing; S.O.: Data Curation, Writing – Review & Editing; M.U.: Funding Acquisition, Project Administration, Supervision, Writing – Review & Editing; K.M.H.: Funding Acquisition, Methodology, Project Administration, Supervision, Writing – Review & Editing; S.B.: Conceptualization, Data Curation, Formal

Analysis, Funding Acquisition, Investigation, Methodology, Project Administration, Resources, Software, Supervision, Validation, Writing – Original Draft Preparation, Writing – Review & Editing.

## Funding

## Competing interests
The authors declare no competing interests.
