## [Peer Review File · Nature Communications]

AI-Based Screening for Thoracic Aortic Aneurysms in Routine Breast MRI

Corresponding Author: Dr Sebastian Bickelhaupt

Version 0:

Reviewer comments:

Reviewer #1

(Remarks to the Author)
Review Nature Communication

Title: AI-Enhanced Breast MRI: Unveiling Hidden Aortic Aneurysms through Routine Dual Screening for Cancer and Cardiovascular Health

Specific Comments:

Abstract: Fine.

Introduction:

Well written. Currently contrast enhanced MRI of the breast is a hot topic (see reference 19). It is hardly to predict if contrast enhanced MRI of the breast will be implemented as a screening tool considering other promising techniques. Nevertheless, it pays off to explore the potential of AI to detect thoracic aortic aneurysm in women who undergo this screening process. Although, the incidence of this disease is very low.

Materials:

The study is IRB approved, multicentric and uses a retrospective study design. A huge dataset (N=5153) is used for the development of the fully automated artificial neural network (ANN) pipeline and analysis. The dataset represents a mix from several study sites in Germany and USA. The dataset represents a mix of several MRI manufacturers and field strengths. This makes the dataset very interesting and robust for any analysis. In accordance to guidelines a training dataset and an independent test dataset has been used to assess the power of the ANN to analyze and detect thoracic aortic aneurysms.

Preprocessing:

The analysis was only performed on one time point after contrast application. It would be interesting to know if measurements of the thoracic aorta are influenced by other post contrast data time points as well.

Segmentation of the thoracic aorta:

Personnel involved:

I would like to know, which expert in which field. In addition, it would be interesting to know which guidelines were used for the segmentation process and how both readers were trained to perform an accurate segmentation. What was the role of the supervisor. How often did he supervise in more than 5000 cases?

Overall, I have to admit that the technique used to train the algorithm is not new, see references.

Nevertheless, I strongly suggest that an expert in AI technologies should review the paper as well.

For the development and assessment of the algorithm a huge dataset was used in a retrospective way. The numbers are impressive and high.

Nevertheless, the algorithm has to be assessed in a prospective huge dataset of multi-institutions as well. In addition, as mentioned in the limitations - the Erlangen dataset - carries too many breast cancer patients and does not reflect a "normal population".

The statistics seems to be correct.

Results:

Overall, nicely presented enhanced with nice figures. As expected, the ANN can perform accurate diameter measurement and can calculate several indices.

As expected, the numbers of diagnosed aneurysm was very low. Of interest would be if any of the diagnosed aneurysm had further assessment with an appropriate CT, angiography or even a treatment. With other words, what would be the consequence of such a diagnosis for any woman?

Based on the given data, the incidence of aortic aneurysm was higher in the breast cancer group. What is the impact of this finding? Are there any explanations for this finding, lifestyle, etc...

Discussion:

Nicely written, some typos, see first page, 3. paragraph, second line.

Lung cancer screening with CT is implemented soon in several countries. Could the proposed algorithm also be used for this study cohort? If so, even with small adaptations, it would be worth to mention that the ANN could be used as a blueprint for other screening programs as well.

In addition, it would be interesting to know how long it takes to perform an analysis per case in seconds or minutes. How could the analysis be included in the reporting.

In addition, the implications of these findings should be discussed as well.

General comment:

In this study the authors demonstrated that a fully automated artificial neural network (ANN) pipeline enables background screening for aortic disease in women who undergo breast MRI. Such a tool is of interest if MRI screening of the breast will be implemented. The study is based on a huge data set from different vendors and different centers in Europe and US, which should be seen as bonus.

The prospective assessment of the ANN in a large cohort is missing and should be seen as a limitation.

The technique used for the ANN is at least for my understanding not unique and has been described in previous papers on smaller cohorts.

(Remarks on code availability)

Reviewer #2

(Remarks to the Author)

This is an interesting study testing whether a deep-learning model can help in identify aortic aneurysms in breast cancer screening MRI. The setting is very relevant, given the low detection rate of aortic aneurysms and their clinical relevance, and the potential impact is vast. The implementation aims at segmenting the thoracic aorta from dynamic contrast enhanced (DCE) T1-weighted (T1w) acquisitions obtained in the context of breast cancer screening, and automatically extract ascending and descending aortic diameters, which are the cornerstone of diagnosis and prognosis of aortic aneurysms. The work has several merits, including a large and diverse dataset and the in-depth analysis of positive cases detected by the algorithm.

Nonetheless, a number of aspects are not sufficiently detailed while certain tests are somehow indirect.

Majors:

- "Analysis of diameters in the aneurysm cases flagged by the ANN compared to manual control segmentations indicated a mean submillimeter deviation of -0.62mm (p=0.92) for the ANN compared to the mean human diameter assessments performed in axial slices" (page 7). Please, provide a measure of dispersion of this error. Moreover, the measurement of aortic diameter on axial slices is suboptimal, especially in regions where the aorta is not perpendicular. This is a limitation that should be reported. More importantly, the authors compared diameters only in studies where the model identified aortic diseases. This is especially problematic given that in this expected clinical context (screening, low pre-test probability and prevalence), the capacity to correctly rule-out aneurysms is particularly relevant. Thus, a test on the negative predictive value is key. Please, consider testing for this important aspect.

- On page 19 the Authors refer to "training dataset was derived from a non-consecutive cohort of n=96 adult patients" but it is unclear how many breast MRI were available. Indeed, few lines down, when introducing the validation data, the Authors refer to "excluding n=217 examinations belonging to the ensembled patients of the training dataset". Please, detail clearly what was the training set.

- "For small segmentation parts of height <25mm, typically resulting from the aortic arch being outside the field-of-view and the ascending aorta being a separate segment" (page 22). It is unclear how it is possible that the remaining length of the

ascending aorta will be <25mm. The ascending aorta is normally 80-120 mm long, so a segment <25 mm would probably consider only the very proximal segment. The prevalence of dilation of the very proximal ascending aorta is much less than of the mid or distal ascending aorta. Please, comment. Moreover, please, clearly state what rule was used to decide where to start the segmentation proximally.

- The section "plane categorization" is unclear (page 23). Normally, anatomic landmarks, particularly the location of the supra-aortic branches, are used for this purpose. Beyond improving the description of this part, the Authors should consider validate (in a small dataset) their plane categorization by manually marking the location of supra-aortic vessels and then checking whether the categorization works correctly. As implemented, it relies heavily on certain assumptions, which may well not hold true in many subjects.

- "3.5-fold improvement of the relative detection rate". While it is a very good result, it is obtained over a very small number of patients, which limits its strength. Please, make sure that an unexperienced reader will understand this limitation.

- Table 2. "Successfully segmenting thoracic aorta" suggest that the algorithm segmented correctly the aorta, a thing that, as far as I understand, it was not tested. As that, the reported feasibility rate is overestimated.

- While the differences in aortic diameter in woman with and without breast cancer is statistically significant the differences are so small (3.05 vs 2.96 cm in the ascending and 2.28 and 2.21 cm in the descending aorta) that are most likely insignificant from a clinical perspective and reasonably smaller than the diameter measurement error. Please, comment.

- Figure 3. How was the ± 7 mm range decided? What is the rationale and implications? It seems arbitrary and very large. Moreover, given the different ranges of values for the ascending and descending aorta, the axes of these scatter plots do not allow to evaluate clearly the associations in the descending aorta. Please, reformulate these scatter plots to provide a clearer view of the results, and consider to add the corresponding Bland-Altman plots.

- I do not understand why the comparison of aortic diameter predicted vs "ground truth" is tested via paired test. Please, explain. I do understand why paired tests are used in the analysis of predictions over consecutive scans.

- Figure 4 should include diameter differences, not their absolute values. Indeed, the whole focus of this section is to test the intra-patient agreement of subsequent predictions. Again, no proper validation is presented for this part: the Authors should consider manually-measuring these diameters and compare them with the predictions.

- "exceeding previously reported Dice coefficients for aortic assessments in dedicated thoracic MRI (0.85)". Many studies reported substantially higher DICE score in the segmentation of the aorta: 10.1007/s00330-022-09068-9; 10.1002/mrm.28257; 10.1002/jmri.27995, for example. Please, reformulate.

- "Inclusion of the ANN in (commonly biennial) breast cancer screening might thus further enable effortless monitoring without additive burden for the patients or costs for the healthcare system by additive examinations". While it is true that there is no consensus on how often patients with aortic aneurysms should be evaluated to monitor disease progression, the suggestion that the present implementation may permit the assessment of it every two years should be contrasted by its unlikely capacity to quantify aortic growth over this period. This is especially relevant as the Authors report a diameter with 2 mm range, and thus a growth of 1 mm/year should be the minimum for the implementation to detect a growth. Please, consider that growth is normally much lower (possibly at around 0.1-0.3 mm/year) and that even current "best" clinical standard fails to detect such limited growth in a two-year time span (10.1007/s00330-021-08273-2).

Minors:

- Computing (and even measuring) the diameter of the aorta in the aortic arch is a very complex task, as the existence of supra-aortic vessels (here were not segmented) poses a special problem in the diameter assessment. On the other hand, the prevalence of aneurysms covering only the arch is very low. I wonder whether the performances on this region are as good as in the ascending and descending aorta, and the percentage of segmentation errors in this region. Please, feel free to disregard this comment, as the paper does not focus on the aortic arch.

- "no combination resulted in a circularity $\geq 85\%$ then the plane is discarded" (page 23). How often does it happen?

- How were "T1w-contrast-enhanced sequences reflecting the early phase (until ~ 120 sec. after GBCA-injection)" identified?

- "screening only offered to high-risk groups such as first-degree relatives" (page 4). Please, underline that "first-degree relatives" is of patients with diagnosed thoracic aortic disease.

- The Authors mix "thoracic aortic disease" with "aneurysms" but thoracic aortic diseases comprise other aortic diseases beyond aneurysms. To avoid overstatements, please, make sure to refer to aneurysms.

- "For the examinations with height/weight available". Please, state the rate of missing values.

- In certain parts of the text the Authors write that Erlangen test dataset had a $n=3232$, but in others (such as page 19) refer to $n=3328$. While the difference is really minimal, please, correct or explain.

- The Authors refer to artificial neural network (ANN) but they use convolution neural network. Please, consider this last name for these networks, which matches more precisely the implemented models.

- "Using this approach might enable effortless screening for a relevant disease otherwise not efficiently addressable by separate screening measures without any additive appointments or medical examinations – if succeeding in reliably providing personalized aortic assessment comparable to e.g. thoracic MRI" (page 5). This sentence is difficult to read: please, consider simplifying it.

- Methods, page 21. I consider that the description of the neural network could be shortened substantially. How were dice and binary cross-entropy losses combined?

- "In order to account for the non-ECG triggered nature of breast MRI examinations the diameters calculated were considered with a 2mm threshold of variance, as previously reported" (page 23). Please, add a reference.

- Line numbering would help in the review process.

- Figure E2. Please, use "descending" instead of "descendens".

- Please, consider using a denser color in Figure 4: it is very difficult to see some dots.

- Typo (page 14): "This might be even more relevant with thoracic aortic disease. This might be even more relevant with thoracic aortic disease"

- The discussion is very long.

- "the real-world robustness of the algorithm" cannot be tested with trial data.

(Remarks on code availability)

Reviewer #3

(Remarks to the Author)

The manuscript presents a fully-automated artificial neural network (ANN) pipeline for detecting thoracic aortic disease using breast MRI. The pipeline involves an initial 3D segmentation of the aorta using nnUNet, followed by diameter calculation for risk assessment. The experiment results demonstrate improvement in aneurysm detection rate compared to traditional clinical readings. However, despite the promising initial results, a few fundamental concerns need to be addressed.

Major Concerns:

1. Clinical implications: The manuscript needs to provide a more thorough explanation regarding the clinical relevance of this approach. MRI is typically used in breast cancer diagnosis following ultrasound and mammography, and it's not a standard screening tool. Additionally, it appears from the BI-RADS categories of the subjects that most participants in this study have some form of breast disease. Therefore, the practical impact of this approach may be limited.
2. In routine MRI scans, some breast MRI may not fully cover all parts of the thoracic aorta, such as the aortic arch. As a result, the reliability of aortic measurements across different breast MRI protocols and imaging centers may be questionable. This further limits the clinical applicability of the proposed AI model.
3. Threshold selection: the manuscript should discuss in detail the selection of thresholds (for risk assessment and determining the presence of aortic disease) and their impact on sensitivity and specificity of detection. It is worth noting that thresholds provided in clinical guidelines may not always yield the best performance in all contexts.
4. The scale of the training dataset: the training dataset only includes MRI data from 96 adult patients without normal subjects. This small-scale sample could potentially limit the generalizability and robustness of the developed framework.
5. If direct aortic diameter measurement can determine risk, why is a broader risk assessment necessary? What is the clinical significance of risk assessment?
6. Lack of prospective validation: While the retrospective analysis shows promising results, the true clinical utility and performance of the proposed framework can only be reliably assessed through prospective trials.

Minor Concerns:

1. Contrast requirement for segmentation: The manuscript needs to clarify whether the proposed segmentation method is limited to contrast-enhanced breast MRI scans, or if it can also be applied to non-contrast (plain) MRI scans.
2. How should cases be handled when the ascending or descending aorta measurements exceed the threshold? Is ultrasound test needed?

Missing details:

Please include details about computational resources for development of the model, such as GPU specifications.

Addressing these concerns will provide a more comprehensive understanding of the methodology and its potential impact on clinical practice.

(Remarks on code availability)

Version 1:

Reviewer comments:

Reviewer #1

(Remarks to the Author)

Review Nature Communication REVISION 1.

Title: AI-Enhanced Breast MRI: Unveiling Hidden Aortic Aneurysms through Routine Dual Screening for Cancer and Cardiovascular Health

This is a resubmission. The authors worked hard to improve the manuscript by addressing all comments. At least for this reviewer the paper improved significantly.

Some minor comments:

Introduction:

Well written but too long. The breast MRI part can be shortened. No one doubts that this is an established method with several indications including screening.

Stability across different post-GBCA-administration:

Chapter hard to read. Shorten your sentences.

Limitations: retrospective data: May I suggest to add arguments as written in the comments for the reviewers

Conclusion of discussion: should be rephrased.

MM: please rephrase: ...Herein, the datasets reflect the diverse settings breast MRI is performed in, in order of strengthening the practical impact of the approach:

(Remarks on code availability)

Reviewer #2

(Remarks to the Author)

During this round of revision, the Authors were able to satisfactorily demonstrate good performance in the few tests, especially aortic diameter quantification and location detection, that were not convincing in the first submission. I particularly appreciate the test on aortic diameter in non-flagged cases, which complements very well the previously-reported results on positive cases and meet the main challenge (rule out false negative).

I still have only minor remarks.

The rate of repeat measurements that fall within 2mm of each other was good (85.55). However, "The ANN measurements demonstrated a high robustness for different timepoints after GBCA injection" seems quite exaggerated. Indeed, the results (Fig 4a) do not look that good: while the measures are quite centered at a very similar mean, the variability is important.

Please, add confidence intervals in figure 4b, second and third lines, and a more balanced discussion of this aspect.

"Significance testing was performed using a t-test with paired samples (AA: $p=0.011/p=0.017$; DA: $p=0.876/p=0.749$; Arch: $p=0.598/p=0.349$, for the Erlangen test set and EA1141, respectively)." Please, reformulate the sentences: neither what was tested nor the meaning of the different p-values are clear.

Please, consider rewriting this sentence: "This workflow integration, which does not require modification of any examination or assessment steps and the ANN therein was found to even demonstrate a slightly lower measurement variation as compared between two radiologists performing >1000 control measurements".

Please, consider rewriting this sentence: "Herein, the datasets reflect the diverse settings breast MRI is performed in, in order of strengthening the practical impact of the approach".

(Remarks on code availability)

Reviewer #3

(Remarks to the Author)

The updated version addresses most of the previous concerns and shows noticeable improvement. However, a few issues still require attention:

1. While the findings indicate enhanced aneurysm detection rates, the possibility of higher false positive rates, which could lead to unnecessary follow-ups and wasted resources. Should be analyzed through additional experiments.
2. It is important to include in the limitations section that some breast MRI scans may not fully image full regions of the thoracic aorta, such as the aortic arch, which may restrict the method's applicability.
3. An ablation study should be conducted to assess how the size of the training dataset affects model performance.

Addressing these points will further strengthen the manuscript.

(Remarks on code availability)

Reviewer #4

(Remarks to the Author)

While most of the concerns raised by Reviewer 3 have been successfully addressed, I still have a few additional points for consideration, which I have outlined below:

1. Reviewer 3 raised a valid concern regarding the generalizability and robustness of the model, particularly due to the small-scale sample used in training. While the authors argue that training the model with 25 cases delivered promising

performance, I still encourage the authors to conduct more rigorous experiments by varying the number of training samples. This would provide a clearer understanding of the model's generalizability and robustness.

2. The authors use the nnU-Net architecture for segmentation, which is a strong choice, but the manuscript lacks sufficient detail regarding the hyperparameter tuning process. Specifically, it is unclear which hyperparameters were tunable, how they were optimized (e.g., learning rate, batch size), and how the model was optimized for this particular task. Given that nnU-Net is highly adaptable, understanding the specific configuration choices made for this task is critical for reproducibility and future model development.

3. Statistical tests are extensively used throughout the evaluation. However, multiple statistical tests are performed across different datasets (Erlangen, Duke, EA1141) and analyses (e.g., segmentation performance, aortic diameter measurements, risk assessment). The manuscript does not explicitly mention whether any corrections for multiple hypothesis testing were applied. Given the potential for inflated Type I error due to multiple comparisons, it is important that the authors address this and provide clarity on whether such corrections were applied, or justify why they were not deemed necessary.

4. While the authors mention performing data augmentation during training (e.g., Rician noise, motion artifacts, and MRI spike artifacts), the manuscript does not explicitly evaluate the model's performance when subjected to noise or artifacts during inference. A dedicated evaluation of the model's robustness to different types and levels of noise or perturbations is necessary to ensure its reliability in clinical practice. This would include testing the model on noisy or perturbed MRI data, which is often encountered in routine clinical settings.

5. The manuscript acknowledges that breast MRI scans were not ECG-gated, which could introduce variability in aortic diameter measurements due to cardiac motion. While the authors note that they accounted for a 2mm variance threshold, this may still affect the accuracy of measurements, especially for smaller aneurysms or those near the diagnostic threshold. It would be beneficial for the authors to discuss the potential limitations of non-ECG-gated imaging in more detail. Could the inclusion of ECG-gated sequences improve measurement accuracy? Additionally, it would be valuable to explore how the ANN might perform when applied to ECG-gated MRI images and whether future work could include this enhancement.

6. There is limited discussion on the interpretability of the model's decision-making process. In clinical applications, understanding the rationale behind a model's predictions is crucial for radiologists and clinicians. The authors should consider incorporating explainability techniques to improve the transparency of the model's decision-making. This would help clinicians understand which aspects of the MRI images contribute to the model's findings and increase confidence in its outputs.

(Remarks on code availability)

Version 2:

Reviewer comments:

Reviewer #3

(Remarks to the Author)

Previous comments were mostly addressed.

(Remarks on code availability)

Reviewer #4

(Remarks to the Author)

While the authors have satisfactorily addressed most of my concerns, a few minor issues remain that I would like to see clarified:

1. Hyperparameters: Given the critical role of hyperparameters in achieving optimal performance in ANNs generally, please elaborate on how nnU-Net determines these values. Are they the result of an optimization process, or are they default settings derived from prior experiments? A brief explanation would be especially useful for readers who are not familiar with the nnU-Net framework.

2. Model's Decision-Making Process: To clarify my previous comment, I am referring specifically to how the model arrives at its segmentation predictions. Although explainability techniques in image classification have received considerable attention, similar methods for semantic segmentation have been relatively neglected. A discussion on this topic, potentially discussing recent surveys such as [1], would enhance the paper.

[1]: Gipiškis, Rokas, Chun-Wei Tsai, and Olga Kurasova. "Explainable AI (XAI) in image segmentation in medicine, industry, and beyond: A survey." ICT Express 2024.

(Remarks on code availability)

Review response for manuscript:

“AI-Enhanced Breast MRI: Unveiling Hidden Aortic Aneurysms through Routine Dual Screening for Cancer and Cardiovascular Health”

We want to sincerely thank the reviewers for their responses and suggestions, as well as for participating in the review process. We believe your feedback has prompted us to majorly improve our work by including:

- a) multiple additive analyses of the >5000 breast MRI datasets as requested by the respective reviewers,
- b) extensive extended manual verification of the ANN results by radiologists as requested,
- c) substantial revision of the manuscript, especially the introduction and discussion, increasing the perspective on the clinical impact, supporting the clarity in the manuscript and fixing typos/errors.

We have carefully considered each point raised and have made the corresponding revisions to the manuscript. Please find below detailed responses for each comment.

Reviewer 1

Specific Comments:

Abstract: Fine.

R1.1. Introduction:

Well written. Currently contrast enhanced MRI of the breast is a hot topic (see reference 19). It is hardly to predict if contrast enhanced MRI of the breast will be implemented as a screening tool considering other promising techniques. Nevertheless, it pays off to explore the potential of AI to detect thoracic aortic aneurysm in women who undergo this screening process. Although, the incidence of this disease is very low.

Answer: We thank the reviewer for the appreciation of our introduction and for participating in the review process. We would like to highlight that the impact of this work is not limited solely to the potential introduction of breast MRI for population based screening (of course this would significantly increase its impact). The diversity of current clinical indications for breast MRI is as well reflected in the different independent breast MRI datasets including screening applications (EA1141) and “typical” other current indications. As breast MRI is already an established imaging modality in certain breast cancer diagnosis pathways and surveillance, identifying incidental aortic diseases in those vulnerable patients might be considered highly relevant even in the present indications of breast MRI. With an estimated up to 2% of MRI examinations being breast MRI (<https://magnetic-resonance.org/ch/21-01.html>) and nearly 40 million MRI examinations per year

in the US, up to 800.000 breast MRI examinations might potentially benefit from such a secondary-data-use approach, which might be an effective way to identify relatively rare but potentially relevant pathologies. We adjusted the introduction and discussion to further highlight the broad potential clinical impact of the approach.

R1.2. Materials:

The study is IRB approved, multicentric and uses a retrospective study design. A huge dataset (N=5153) is used for the development of the fully automated artificial neural network (ANN) pipeline and analysis. The dataset represents a mix from several study sites in Germany and USA. The dataset represents a mix of several MRI manufacturers and field strengths. This makes the dataset very interesting and robust for any analysis. In accordance to guidelines a training dataset and an independent test dataset has been used to assess the power of the ANN to analyze and detect thoracic aortic aneurysms.

Answer: We thank the reviewer for the appreciation of our methodological approach. Indeed, providing this huge multi-institutional dataset with breast MRI data from different vendors, field strengths, and MR imaging protocols was a very sizable effort and we are grateful it gets recognized.

R1.3. Preprocessing:

The analysis was only performed on one time point after contrast application. It would be interesting to know if measurements of the thoracic aorta are influenced by other post contrast data time points as well.

Answer: We agree that this is a relevant aspect. We selected early time points to maximize the potential applicability of the approach to different breast MRI protocol settings. That is because abbreviated protocols are of increasing interest and might only acquire a single (early) time point after contrast agent injection. We amended this in the manuscript now.

*Further, since we agree that the effect of time point selection might be interesting to the reader, we performed an **additive analysis** and now provide **additive insight on the generalizability on different timepoints after contrast agent administration for the >3000 breast MRIs from the Erlangen test dataset** as part of our revision (the breast MRI dataset for which is we consistently have the other time points and they are acquired in a highly standardized manner). We added the analysis in the manuscript and now provide an additive figure as Figure 4b. Our analysis indicated that using later subsequent timepoints does not significantly influence performance of the ANN.*

R1.4. Segmentation of the thoracic aorta:

Personnel involved:

I would like to know, which expert in which field. In addition, it would be interesting to know which guidelines were used for the segmentation process and how both readers were trained to perform an accurate segmentation. What was the role of the supervisor. How often did he supervise in more than 5000 cases?

Answer: We have now specified the expertise in the field of imaging for the personnel involved in segmentation of ground truth data – in short research assistants (medical students) supervised by board certified radiologists (now with given sub-speciality experience) provided the training data and the evaluations were then conducted by board certified radiologists only. We included the guidelines and references which were used for the image segmentation process and now more clearly mark them in the manuscript.

Given the comments of the second reviewer we performed an extensive additive evaluation of the data:

- *We performed a **full validation for all of the >5000 breast MRI examinations** as the following (and now included in the methods):*
 - *All cases of the independent test datasets were manually validated independently by two board-certified radiologists with regards to segmentation accuracy and plane selection appropriateness. This was done to ensure appropriate segmentation determination and NPV of the algorithm in the independent test datasets.*
- *Further we randomly performed **>1000 manual diameters measurements in the ascending and descending aorta to determine precision of the algorithm** in evaluating the diameter of the respective aortic section. Diameter measurements performed by both of the two board-certified radiologists independently and blinded to the diameter of the ANN.*

Thus, while occupying multiple days, evenings and weekends we indeed evaluated all the cases by the two involved board certified radiologists, which we believe has greatly improved the quality of the manuscript.

R1.5. Overall, I have to admit that the technique used to train the algorithm is not new, see references.

Nevertheless, I strongly suggest that an expert in AI technologies should review the paper as well.

Answer: Yes, indeed the used AI technique for deriving the segmentations itself was not completely new, apart from the extended augmentation scheme. However, the quiet complex algorithmic measurement of the aortic diameters and the plane categorization, as well as the

creation of the reporting document for the purpose of this study, also contributed to the methodological novelty of the work.

Building upon a robust and extensively validated AI framework for segmentation purposes allowed us to derive this innovative approach in breast MRI and it is something that we consider a strength with regards to scientific reproducibility and robustness of the presented approach. We used a highly-established neural network approach for this, aiming to avoid the existing so-called “innovation bias towards novel [AI] architectures” as recently described^[BS1]. Further, we believe we contribute through our extensive evaluation, hoping to set a standard on how future methods should be evaluated.

[BS1]<https://arxiv.org/pdf/2404.09556>

R1.6. For the development and assessment of the algorithm a huge dataset was used in a retrospective way. The numbers are impressive and high.

Nevertheless, the algorithm has to be assessed in a prospective huge dataset of multi-institutions as well. In addition, as mentioned in the limitations - the Erlangen dataset - carries too many breast cancer patients and does not reflect a “normal population”.

The statistics seems to be correct.

Answer: We thank the reviewer for appreciating the diverse, multi-site, multi-vendor, multi-field strength, multi-protocol approach of our evaluation! We address the topic of prospective evaluation later in the response (R1.13). We would like to also thank the reviewer for the verification of our statistical analysis.

R1.7. Results:

Overall, nicely presented enhanced with nice figures. As expected, the ANN can perform accurate diameter measurement and can calculate several indices.

As expected, the numbers of diagnosed aneurysm was very low. Of interest would be if any of the diagnosed aneurysm had further assessment with an appropriate CT, angiography or even a treatment. With other words, what would be the consequence of such a diagnosis for any woman?

*Answer: This is indeed a relevant aspect: **We have now added a case description visualizing the breast MRI and its context as well as the subsequent clarification process with CT angiography and related findings. We added this information in the Extended data figure section (Extended data figure E3).***

R1.8. Based on the given data, the incidence of aortic aneurysm was higher in the breast cancer group. What is the impact of this finding? Are there any explanations for this finding, lifestyle, etc...

*Answer: We thank the reviewer for commenting on this highly relevant aspect. **We expanded our analysis of this correlation and performed additive evaluations.** When including not only women with current breast cancer but as well those with a history of breast cancer the correlation and OR further increased in our study. There is indeed some limited literature that aortic aneurysms might grow faster in cancer patients receiving a chemotherapy treatment (Becker von Rose A, Kobus K, Bohmann B, Lindquist-Lilljequist M, Eilenberg W, Bassermann F, Reeps C, Eckstein HH, Trenner M, Maegdefessel L, Neumayer C, Brostjan C, Roy J, Hultgren R, Schwaiger BJ, Busch A. Radiation and Chemotherapy are Associated with Altered Aortic Aneurysm Growth in Patients with Cancer: Impact of Synchronous Cancer and Aortic Aneurysm. *Eur J Vasc Endovasc Surg.* 2022 Aug-Sep;64(2-3):255-264. doi: 10.1016/j.ejvs.2022.07.007. Epub 2022 Jul 16. PMID: 35853577.) however conflicting results (even from the same group) exist: Kobus K, Bohmann B, Wilbring M, Kapalla M, Eckstein HH, Bassermann F, Stratmann JA, Wahida A, Reeps C, Schwaiger BJ, Busch A, von Rose AB. Cancer, cancer treatment and aneurysmatic ascending aorta growth within a retrospective single center study. *Vasa.* 2023 Jan;52(1):38-45. doi: 10.1024/0301-1526/a001038. Epub 2022 Nov 14. PMID: 36373268.*

*Further there is some literature that patients with aortic aneurysms might be at elevated risk for cancer (Cancer Incidence After Diagnosis of Abdominal Aortic Aneurysm—Brief Report; L. Luo, A. M. Haas, C. F. Bell, R. A. Baylis, S. S. Adkar, C. Fu, et al.; *Arteriosclerosis, Thrombosis, and Vascular Biology* 2024 Vol. 44 Issue 7 Pages 1694-1701, DOI: doi:10.1161/ATVBAHA.123.320543,*

<https://www.ahajournals.org/doi/abs/10.1161/ATVBAHA.123.320543>

Given however, that especially breast cancer chemotherapy is sometimes associated with relevant cardiotoxicity it might be reasonable that the associated aorta (receiving a relatively high concentration of chemotherapy due to its central cardiovascular position) might be affected by the treatment as well. Our study is of correlative character and we cannot clear up the causal reason for the finding (that would require a study on the national/international level), however the secondary implications from our results are important: Many women receiving breast MRI today are breast cancer patients and/or survivors. If - as the data suggests - they indeed seem to possess an elevated risk for presenting aortic diseases and the data to screen for it is already at hand but currently not adequately assessed in clinical routine, the approach we presented is of even higher importance since it would address a population at specific risk in this disease spectrum.

R1.9. Discussion:

Nicely written, some typos, see first page, 3. paragraph, second line.

Answer: We thank the reviewer for pointing to the typos. We have addressed them.

R1.10. Lung cancer screening with CT is implemented soon in several countries. Could the proposed algorithm also be used for this study cohort? If so, even with small adaptations, it would be worth to mention that the ANN could be used as a blueprint for other screening programs as well.

Answer: Yes indeed, we expanded on that in our discussion.

R1.11. In addition, it would be interesting to know how long it takes to perform an analysis per case in seconds or minutes. How could the analysis be included in the reporting. In addition, the implications of these findings should be discussed as well.

*Answer: Our approach can run **completely independently in the background** and produce a PDF “lab report” for the reading radiologist to see during the examination reading (refer to Extended data figure E2 for an example), which would very prominently highlight any dilation. Due to this, the actual visual evaluation itself by the clinician can be done within a couple of seconds. We did not measure in-depth the actual software processing time to produce the report, as it is (i) less relevant, given that it runs completely independently in the background and can fetch data directly from the hospital PACS, making it immensely faster than the typical time between scan acquisition and radiologist reading and (ii) the time-to-process is very hardware-dependent. Nonetheless, in our hardware, processing a case takes around 1-8 minutes (depending on the image size and the number of computationally-expensive rotations needed) – however the process is highly parallelizable, so many cases can be processed in parallel, making the actual time needed a lot less.*

Regarding how this analysis could be included in the process, in our discussion we address it in this part: “The ANN is capable of running fully automatically in the background in a clinical routine workflow providing a structured report”. We do not envision any complex workflows - the radiologist who performs the examination readings can have the results of the ANN pipeline available in PDF or similar format to aid in diagnosis.

R1.12. General comment:

In this study the authors demonstrated that a fully automated artificial neural network (ANN) pipeline enables background screening for aortic disease in women who undergo breast MRI. Such a tool is of interest if MRI screening of the breast will be implemented. The study is based on a huge data set from different vendors and different centers in Europe and US, which should be seen as bonus.

Answer: We thank the reviewer for the appreciation of our study.

R1.13. The prospective assessment of the ANN in a large cohort is missing and should be seen as a limitation.

Answer: Yes, indeed a prospective cohort has not been part of the study and we mention that already in our limitations.

Whilst we agree that a prospective multicentric randomized clinical trial can be considered gold standard, we are addressing a topic in which the status “disease” is mostly defined by variation of the diameter. The structure itself (aorta) is present in every single patient, thus successfully determining a diameter in the aorta can be seen as potentially indicating the robustness in finding diseases as well.

*As we would need probably **years of prospective studies** with again thousands of new examinations (due to the relatively rare disease) we believe there is benefit to a retrospective approach which includes **different multisite studies, data from all major vendors and clinical field strengths**, which can be seen as a indicator of high robustness - despite of course a prospective clinical trial always being the gold standard in evidence based medicine.*

*It is also worth noting that despite its retrospective nature, our dataset from the University Hospital was consecutive, **comprising all examinations performed in the premises within a 5 year period**, laying a strong foundation that successful prospective evaluation is feasible and justifying the likely costly multi-year endeavor now ahead of us.*

*However, we assume that in a prospective clinical study (in which the radiologists would be aware that in parallel to them an AI is searching for aortic diseases) **we might even potentially introduce a diagnostic awareness bias limiting real world insights as compared to our current evaluation approach** in which the radiologists reports were usable as real world indicators of reporting frequency of aortic diseases. We have now expanded on this in the manuscript’s limitations and we hope our response suits the reviewer’s comment accordingly.*

R1.14. The technique used for the ANN is at least for my understanding not unique and has been described in previous papers on smaller cohorts.

*This is partially true: indeed the used AI technique for deriving the segmentations itself was not completely new, other than the algorithmic measurement of the aortic diameters and the creation of the reporting document for the purpose of this study. Indeed, building upon a robust and extensively validated AI for segmentation purposes allowing to derive this innovative approach for breast MRI is something that we consider a strength with regards to scientific reproducibility and robustness of the approach. Using a highly established, award winning neural network approach was further aiming to avoid the highly relevant so-called “**innovation bias towards novel architectures**” as recently described^[BS1] as a severe limitation of ever “new” AI algorithms in diagnostic imaging.*

[BS1] <https://arxiv.org/pdf/2404.09556>

Reviewer 2

This is an interesting study testing whether a deep-learning model can help in identify aortic aneurysms in breast cancer screening MRI. The setting is very relevant, given the low detection rate of aortic aneurysms and their clinical relevance, and the potential impact is vast. The implementation aims at segmenting the thoracic aorta from dynamic contrast enhanced (DCE) T1-weighted (T1w) acquisitions obtained in the context of breast cancer screening, and automatically extract ascending and descending aortic diameters, which are the cornerstone of diagnosis and prognosis of aortic aneurysms. The work has several merits, including a large and diverse dataset and the in-depth analysis of positive cases detected by the algorithm.

Nonetheless, a number of aspects are not sufficiently detailed while certain tests are somehow indirect.

Answer: We thank the reviewer for the appreciation of our work and for participating in the review process. We have significantly expanded our work and we hope their concerns are addressed.

Majors:

R2.1.- “Analysis of diameters in the aneurysm cases flagged by the ANN compared to manual control segmentations indicated a mean submillimeter deviation of -0.62mm ($p=0.92$) for the ANN compared to the mean human diameter assessments performed in axial slices” (page 7). Please, provide a measure of dispersion of this error.

R2.2. Moreover, the measurement of aortic diameter on axial slices is suboptimal, especially in regions where the aorta is not perpendicular. This is a limitation that should be reported.

R2.3 More importantly, the authors compared diameters only in studies where the model identified aortic diseases. This is especially problematic given that in this expected clinical context (screening, low pre-test probability and prevalence), the capacity to correctly rule-out aneurysms is particularly relevant. Thus, a test on the negative predictive value is key. Please, consider testing for this important aspect.

Answer: We thank the reviewer for raising these important aspects (R2.1-R2.3). Accordingly, we significantly expanded our work (as well referring to the comments for reviewer 1).

*We performed a **full validation** for **all of the >5000 breast MRI examinations** as the following (and now included description in the methods and results in the results section/figures/tables):*

- *All cases of the independent test datasets were **manually validated independently by two board-certified radiologists with regards to segmentation accuracy and plane selection appropriateness**. This was done to ensure appropriate segmentation*

determination and NPV of the algorithm in the independent test datasets (addressing comments R2.2 and R2.3).

Further we randomly performed >1000 manual control diameters measurements in the ascending and descending aorta to determine precision of the algorithm in evaluating the diameter of the respective aortic section with two radiologists. Dispersion of error is now as well added and substantiated significantly by the additive measurements.

Additively we repeated as well all the measurements of all cases of aneurysms and expanded it to all dilatation cases for the datasets (40mm and higher and 30mm and higher) in angulated planes to provide this robust data, despite our assumption that in clinical routine most likely the images might be rather axially analyzed. Interestingly, this reduced the average deviation of the ANN derived diameters to the manually derived diameters by about 20%.

Thus, while occupying multiple days, evenings and weekends we indeed evaluated all the cases by the two involved board certified radiologists, which we believe has greatly improved the quality of the manuscript.

R2.4- On page 19 the Authors refer to “training dataset was derived from a non-consecutive cohort of n=96 adult patients” but it is unclear how many breast MRI were available. Indeed, few lines down, when introducing the validation data, the Authors refer to “excluding n=217 examinations belonging to the ensembled patients of the training dataset”. Please, detail clearly what was the training set.

Answer: We understand where the confusion stems from and we have further clarified in the manuscript. There were n=96 examinations from n=96 patients included in the training set. However, a total of n=217 examinations (n=96 used for training and n=121 examinations belonging to the patients used for training but acquired on different dates) had to be excluded from the test set, in order to ensure a fair evaluation. We clarified this further in the text.

R2.5. “For small segmentation parts of height <25mm, typically resulting from the aortic arch being outside the field-of-view and the ascending aorta being a separate segment” (page 22). It is unclear how it is possible that the remaining length of the ascending aorta will be <25mm. The ascending aorta is normally 80-120 mm long, so a segment <25 mm would probably consider only the very proximal segment. The prevalence of dilation of the very proximal ascending aorta is much less than of the mid or distal ascending aorta. Please, comment. Moreover, please, clearly state what rule was used to decide where to start the segmentation proximally.

Answer: This was obviously phrased in a way that welcomes misunderstandings - we apologize for that. What we meant is that the algorithm additively considered rare cases, wherein only a very small part of the ascending aorta was actually visible on the MRI, due to the acquisition position, and thus only analyzed that segment.

Segmentation was started at the breast MRI slice which provided the first visual clear depiction of the ascending aorta, which commonly was either the bulbous or latest the sinotubular junction section. We rephrased accordingly in the manuscript to make this decision rule more clear to the reader.

R2.6. - The section “plane categorization” is unclear (page 23). Normally, anatomic landmarks, particularly the location of the supra-aortic branches, are used for this purpose. Beyond improving the description of this part, the Authors should consider validate (in a small dataset) their plane categorization by manually marking the location of supra-aortic vessels and then checking whether the categorization works correctly. As implemented, it relies heavily on certain assumptions, which may well not hold true in many subjects.

Answer: We thank the reviewer for raising this important aspect. We agree that for analyzing the aorta in breast MRI acquisitions some technical compromises were necessary. As a breast MRI acquisition is not intended for full thoracic aorta depiction and routinely ends “somewhere” in the aortic arch area, we needed to find an alternative to factually detecting the supraaortic branches.

*We **validate the plane categorization in a subset of n=50 samples manually now**. Still it is important to us to clarify that a breast MRI examination is not equivalent to a dedicated thoracic aortic MRI. This inevitable limitation of “aortic-incomplete” datasets will always be there – still our approach demonstrated how many more cases of aortic disease could be identified, even with this inherent limitation of assessing the aorta in breast MRI examinations. We included the results as well as the methods for validation of plane categorization now in the manuscript.*

R2.7- “3.5-fold improvement of the relative detection rate”. While it is a very good result, it is obtained over a very small number of patients, which limits its strength. Please, make sure that an unexperienced reader will understand this limitation.

Answer: We thank the reviewer for raising this important aspect. Accordingly, we emphasized this aspect now already at the very beginning of the discussion and again in the limitations section.

R2.8- Table 2. “Successfully segmenting thoracic aorta” suggest that the algorithm segmented correctly the aorta, a thing that, as far as I understand, it was not tested. As that, the reported feasibility rate is overestimated.

Answer: We agree with the reviewer that “successfully segmented” was not the correct term - the reported metric was how often the machine learning model successfully provided a measurement from the respective aorta sections. To address this, we significantly expanded our work as stated above.

We now included the details on error rates and sources in the manuscript. As described in the manuscript, we considered the AI to be not at fault in cases where e.g. no segmentation was possible due to a missing depiction associated with the breast MRI acquisition (e.g. chosen FoV). We corrected the wording, performed additive extensive data analysis and give more insights now in Table 2 as stated above by the extensive additive analysis.

R2.9- While the differences in aortic diameter in woman with and without breast cancer is statistically significant the differences are so small (3.05 vs 2.96 cm in the ascending and 2.28 and 2.21 cm in the descending aorta) that are most likely insignificant from a clinical perspective and reasonably smaller than the diameter measurement error. Please, comment.

Answer: We thank the reviewer for this comment. Indeed, the differences in absolute mm are relatively small – however, the two distributions (refer to Extended data figure E3) are significantly different and the distribution of diameters for patients with breast cancer is quite clearly “shifted” towards higher values, which we would not expect to be attributed to measurement error (which would have added noise in all directions).

Further, we also expanded the group selection to provide more insight into this correlation. This showed that considering breast cancer history (alongside a current diagnosis breast cancer) further (slightly) increased the difference in between the groups.

Whilst our dataset is not suited to provide long-term predictive insight, it might still be highly relevant to decipher such subtle patterns for future prognostics works, as well as help justify larger national/international-level studies on the topic. Despite the marginally appearing differences, the significance might not be irrelevant as associated with a very current publication: Therein for computed tomography imaging (CT) it was shown that amongst the ten most relevant risk factors predicting the 10 year mortality the assessment of the thoracic aorta was ranked as the second most relevant factor <https://pubs.rsna.org/doi/10.1148/radiol.240541>.

We now more extensively discuss this and we include as well this new publication. We hope this addresses the topic raised by the reviewer.

R2.10 Figure 3. How was the $\pm 7\text{mm}$ range decided? What is the rationale and implications? It seems arbitrary and very large. Moreover, given the different ranges of values for the ascending and descending aorta, the axes of these scatter plots do not allow to evaluate clearly the associations in the descending aorta. Please, reformulate these scatter plots to provide a clearer view of the results, and consider to add the corresponding Bland-Altman plots.

Answer: When developing this approach, we had in mind that the radiologists in practice would want some buffer for the aortas that are flagged by the system as dilated (e.g. so that e.g. an ascending aorta with 39.4mm is not considered normal, while 39.6mm - rounded to 40mm - is flagged). We had chosen 2mm as this buffer and added another 5mm for security reasons. As such, 7mm signified the threshold for which an aneurysm - despite potential underestimation of its diameter - is still flagged to be presented to the radiologist. However, we agree that this logic

did not really make it into the manuscript and we have adjusted the threshold to be 5mm in the plots, to signify the threshold that the potentially relevant aneurysms are at least flagged as ectasia and not completely missed.

Further, we thank the reviewer for their suggestion to readjust the plots. We had initially kept all parts of the aorta (ascending, descending, arch) at the same scale so that the differences in size between them are more visible. We agree though that it is beneficial to zoom in a bit for a more clear viewing and we have adjusted accordingly.

We also added Bland-Altman plots as requested.

R2.11 - I do not understand why the comparison of aortic diameter predicted vs “ground truth” is tested via paired test. Please, explain. I do understand why paired tests are used in the analysis of predictions over consecutive scans.

Answer: For the comparison in question, we test two sets for significance: (i) the predicted diameters and (ii) the ground truth diameters. For each individual patient a predicted and a ground truth diameter exists.

We believe that the paired approach might thus be considered appropriate here, however we are happy and open to feedback and kindly invite the reviewer to propose a different approach if they find it more fitting.

R2.12- Figure 4 should include diameter differences, not their absolute values. Indeed, the whole focus of this section is to test the intra-patient agreement of subsequent predictions. Again, no proper validation is presented for this part: the Authors should consider manually-measuring these diameters and compare them with the predictions.

*Answer: We thank the reviewer for their suggestion. We believe the scatterplots are also informative, as they embed information about the diameter differences (which is the distance of a point from the middle line of the plot), but also allow visualizing more intricate details, such as exactly where these differences might be bigger and the overall distribution. **We have added bland-altman plots next to them as requested now.***

*We only intended this analysis to be an additional indirect measure of intra-patient performance and we labeled it as such. However, we agree on the comment that manual-measuring would be appropriate. **To address this, we provide more direct evidence of the quality of the predictions, we performed analysis of >1000 manual aortic diameter control measurements performed by two radiologists and additively now in all ectasia and aneurysm cases and now present the results in comparison to the automatic measurements now in the manuscript.***

R2.13- “exceeding previously reported Dice coefficients for aortic assessments in dedicated thoracic MRI (0.85)”. Many studies reported substantially higher DICE score in the segmentation

of the aorta: 10.1007/s00330-022-09068-9; 10.1002/mrm.28257; 10.1002/jmri.27995, for example. Please, reformulate.

Answer: We thank the reviewer for the comment. Indeed, we wanted to refer to “non”-dedicated MRI studies, which are not primarily intended to allow cardiac assessments (as analysis of the aorta in dedicated heart examinations is of course possible). We now clarified this in the sentence and included the other references for clarity as well.

R2.14- “Inclusion of the ANN in (commonly biennial) breast cancer screening might thus further enable effortless monitoring without additive burden for the patients or costs for the healthcare system by additive examinations”. While it is true that there is no consensus on how often patients with aortic aneurysms should be evaluated to monitor disease progression, the suggestion that the present implementation may permit the assessment of it every two years should be contrasted by its unlikely capacity to quantify aortic growth over this period. This is especially relevant as the Authors report a diameter with 2 mm range, and thus a growth of 1 mm/year should be the minimum for the implementation to detect a growth. Please, consider that growth is normally much lower (possibly at around 0.1-0.3 mm/year) and that even current “best” clinical standard fails to detect such limited growth in a two-year time span (10.1007/s00330-021-08273-2).

*Answer: We fully agree. Indeed, we wanted to hint towards the principle potential of such secondary data use not only for prevalence screening but as well for surveillance. However, as correctly stated, the (luckily) low growth rates are small, **we rephrased the section accordingly to avoid overstatements and hope this is fine for the reviewer.***

Minors:

R2.16- Computing (and even measuring) the diameter of the aorta in the aortic arch is a very complex task, as the existence of supra-aortic vessels (here were not segmented) poses a special problem in the diameter assessment. On the other hand, the prevalence of aneurysms covering only the arch is very low. I wonder whether the performances on this region are as good as in the ascending and descending aorta, and the percentage of segmentation errors in this region. Please, feel free to disregard this comment, as the paper does not focus on the aortic arch.

Answer: We did not find major differences in the performance on the aortic arch, apart from the fact that it is frequently incompletely depicted on the breast MRI examinations due to it being outside the field-of-view. We refrained from making claims on finding aneurysms there, as indeed we do not segment the supra-aortic vessels, as well as other factors: (i) arch aneurysms are rare, as stated, (ii) thresholds for women were not stated in the AHA guidelines, as solid as the ones for the ascending and descending aorta.

R2.17- “no combination resulted in a circularity $\geq 85\%$ then the plane is discarded” (page 23). How often does it happen?

Answer: The circularity threshold was a safety mechanism, because when working with points and their normal vectors (i.e., the defined plane) there can be inaccuracies due to the digital nature of the acquisitions and, most importantly, in acquisitions where the aortic arch is partially present. In those cases we want to discard the planes to avoid miscalculations. Due to this, we took advantage of the tubular structure of the aorta to catch mistakes. We reran all calculations and found that 14.2% of the individual planes (not the entire analyses of the respective breast MRI case) were discarded during processing.

R2.18- How were “T1w-contrast-enhanced sequences reflecting the early phase (until ~120 sec. after GBCA-injection)” identified?

Answer: For the in-house data, we knew the acquisition times from our breast MRI protocol naturally. For the DUKE dataset, a similar process was followed, however we agree that we did not have full insight into the acquisition protocol and can only derive this metric by assuming that basic principles of breast MRI were followed. We include this limitation now in the methods section for the readers. The EA1141 dataset was an abbreviated MRI dataset, so there was a contrast-enhanced acquisitions at 60-90 sec after injection as per study protocol - the major difficulty, which took a lot of time, was identifying which was the contrast-enhanced acquisition for each patient due to the very diverse sources of the data, but we created source code which covers everything (aorta_aneurysm/breast_mri_ops/pick_dicom_series.py).

As it was requested by reviewer 1 we performed and added an evaluation on the different timepoints after GBCA administration now and demonstrate the limited influence of this selection.

R2.19- “screening only offered to high-risk groups such as first-degree relatives” (page 4). Please, underline that “first-degree relatives” is of patients with diagnosed thoracic aortic disease.

Answer: Now included.

R2.20

- The Authors mix “thoracic aortic disease” with “aneurysms” but thoracic aortic diseases comprise other aortic diseases beyond aneurysms. To avoid overstatements, please, make sure to refer to aneurysms.

Answer: Now corrected.

R2.21- “For the examinations with height/weight available”. Please, state the rate of missing values.

Answer: We politely comment that the number of examinations with height/weight was available already on Figure 2b (n_{hw}).

R2.22- In certain parts of the text the Authors write that Erlangen test dataset had a $n=3232$, but in others (such as page 19) refer to $n=3328$. While the difference is really minimal, please, correct or explain.

Answer: There were 3328 examinations used from the Erlangen University Hospital, of which 96 were used for training purposes and 3232 used for evaluation. The total (3328) is only reported in the “Study design and participants” section, for which all patients have to be reported, while when we refer to the “Erlangen test set” we refer only to the $n=3232$ subset of independent test cases which were kept separately for testing after the model development.

R2.23- The Authors refer to artificial neural network (ANN) but they use convolution neural network. Please, consider this last name for these networks, which matches more precisely the implemented models.

Answer: We use the term Artificial Neural Network (in the broad sense, which encompasses Convolutional Neural Networks and other architectures) to make the work more accessible to larger audiences, especially radiologists and other medical practitioners, which we do not expect to necessarily know or care what a CNN is. However, the framework used (nnU-Net) is of course a CNN, so we amended the “Training of the ANN” section to explicitly state that.

R2.24- “Using this approach might enable effortless screening for a relevant disease otherwise not efficiently addressable by separate screening measures without any additive appointments or medical examinations – if succeeding in reliably providing personalized aortic assessment comparable to e.g. thoracic MRI” (page 5). This sentence is difficult to read: please, consider simplifying it.

Answer: We rephrased the sentence accordingly.

R2.25- Methods, page 21. I consider that the description of the neural network could be shortened substantially. How were dice and binary cross-entropy losses combined?

Answer: We tried to include a brief explanation about how nnU-Net works, to provide a bit of context to people that have some general computer science background. However, we agree that the explanation of U-Nets was excessive and such information can generally be found in other sources. As such, we substantially shortened the beginning of the paragraph, to remove the explanation of how U-Nets work. For similar reasons, we decided to not expand on how dice and binary CE losses were combined, as this is a popular approach and the default for nnU-Net; we mostly wanted to highlight that we did not use something else. We however decided to keep all our further described network hyperparameters and our extensions to the augmentation scheme, as this ensures reproducibility and is part of our contribution.

R2.26- "In order to account for the non-ECG triggered nature of breast MRI examinations the diameters calculated were considered with a 2mm threshold of variance, as previously reported" (page 23). Please, add a reference.

Answer: Done

- Line numbering would help in the review process.

Answer: ok

R2.27- Figure E2. Please, use "descending" instead of "descendens".

Answer: We have now changed the text in the figure to "descending".

R2.28- Please, consider using a denser color in Figure 4: it is very difficult to see some dots.

Answer: We had lighter colors for the dots so that they become darker when they overlap, in order to signify higher concentration of dots in some areas. We understand though that the presentation can benefit from some improvement, so we darkened the dots a bit, while making the background shades a bit lighter to improve contrast. Additionally, in the previous analysis we rounded all diameters to the nearest millimeter, to streamline the evaluation and avoid rounding discrepancies. However, we now made the plot without rounding, so it leads to less overlap between dots.

R2.29- Typo (page 14): "This might be even more relevant with thoracic aortic disease. This might be even more relevant with thoracic aortic disease"

Answer: We thank the reviewer for pointing this out. We fixed the text.

R2.30- The discussion is very long.

Answer: We now rephrased some sections in the discussion to shorten the text.

R2.31- "the real-world robustness of the algorithm" cannot be tested with trial data.

Answer: We understand the comment and we have removed "real-world" from this sentence.

Reviewer 3

The manuscript presents a fully-automated artificial neural network (ANN) pipeline for detecting thoracic aortic disease using breast MRI. The pipeline involves an initial 3D segmentation of the aorta using nnUNet, followed by diameter calculation for risk assessment. The experiment results demonstrate improvement in aneurysm detection rate compared to traditional clinical readings. However, despite the promising initial results, a few fundamental concerns need to be addressed.

Answer: We thank the reviewer for the time and work invested in our manuscript. We worked substantially on the manuscript, as well as performed additive evaluation and analyses. We hope that we have addressed all questions raised.

Major Concerns:

R3.1. Clinical implications: The manuscript needs to provide a more thorough explanation regarding the clinical relevance of this approach. MRI is typically used in breast cancer diagnosis following ultrasound and mammography, and it's not a standard screening tool. Additionally, it appears from the BI-RADS categories of the subjects that most participants in this study have some form of breast disease. Therefore, the practical impact of this approach may be limited.

Answer: We thank the reviewer for this comment. Indeed, with our manuscript, we did not aim to suggest the usage of breast MRI for the sole sake of thoracic aortic aneurysm screening. We hope with our revision we clarify any potential misunderstanding. The underlying idea is to better exploit any routine breast MRI data (for whatever reason the breast MRI was indicated in first place, if done after mammography, or as primary screening etc.) in order to more reliably detect concomitant thoracic aortic aneurysms as a potentially highly relevant incidental finding.

*This practical impact is thus independent of the source of the breast MRI data. Whenever a breast MRI is done, as per our study results, using the ANN for aneurysm detection improves the probability that a concomitant aneurysm depicted on the MRI is actually detected. **This practical impact is a well reflected in our data as the datasets specifically reflect the diverse settings breast MRI is performed in: The Erlangen test dataset include the clinical routine variety of clinical indications for breast MRI (high-risk screening, (pre-operative) staging, follow-up, surveillance) over a longitudinal cohort covering 5 years, the DUKE dataset includes breast cancer patients with known disease only and EA1141 includes a cohort undergoing breast MRI screening in women with dense breasts.***

*Importantly, we want to re-emphasize that the Erlangen test dataset with >3000 breast MRI examinations from 5 years reflects a consecutive routine clinical cohort from a large center with the vast majority of the included women(n=~2500) actually **not** having breast cancer.*

We now significantly expanded on this practical impact on our manuscript and hope it addresses the aspects of the reviewer's comment sufficiently.

Lastly, we also now further reference and discuss a recent publication, performed in thoracic CTs, indicating that amongst the ten most predictive factors for 10-year mortality, assessments of the thoracic aorta is ranked second. So we hope this further hints towards the clinical relevance of our findings, given that such a crucial risk factor can effortlessly be evaluated in data that is acquired anyhow and currently just not explored for this aspect.

R3.2. In routine MRI scans, some breast MRI may not fully cover all parts of the thoracic aorta, such as the aortic arch. As a result, the reliability of aortic measurements across different breast MRI protocols and imaging centers may be questionable. This further limits the clinical applicability of the proposed AI model.

Answer: We thank the reviewer for this comment.

We agree on the statement that organ regions which are not captured during the MR examination cannot be analyzed - this is a given intrinsic factor for all secondary screening approaches that use imaging data acquired for a different primary purpose.

We independently tested the reliability across different breast MRI protocols and imaging centers in >5000 breast MR examinations. The datasets included reflect all major MRI vendors, different scanner generations, field strengths etc. So we believe we fairly tried to maximize demonstrating the reliability of the aortic measurements across different settings.

Table two demonstrates that despite this variability, the ANN was pretty robust and successfully determined diameters in the vast majority of examinations (in >90% of all breast MRI examinations). But, we fully agree and stated as well transparently, that it will not work on 100% of all examinations.

We further highlight this potential limitation in the manuscript's limitations section now and with the extended data analysis provide more insight into the underlying reasons for failed ANN cases.

R3.3. Threshold selection: the manuscript should discuss in detail the selection of thresholds (for risk assessment and determining the presence of aortic disease) and their impact on sensitivity and specificity of detection. It is worth noting that thresholds provided in clinical guidelines may not always yield the best performance in all contexts.

Answer: We agree that threshold selection is crucial and that this will affect downstream results. As described in our manuscript we based the thresholds on the current AHA guidelines as a fairly relevant society.

*In order to evaluate different thresholds we **now performed an additive analysis** included in the manuscript, **evaluating different more cautious thresholding approaches** starting already at the dilatation threshold and how this would have influenced the PPV / recall rate in our cohort.*

We hope this new evaluation which is now included in the methods and results addresses the points raised by the reviewer.

R3.4. The scale of the training dataset: the training dataset only includes MRI data from 96 adult patients without normal subjects. This small-scale sample could potentially limit the generalizability and robustness of the developed framework.

Answer:

Regarding dataset composition: The model was indeed trained on patients undergoing breast MRI (assuming this is the non-normality of the patients indicated- we apologize if something else was meant), thus inevitably having a clinical indication for undergoing breast MRI. We are not aware of literature indicating differences in aorta anatomy between women with clinically indicated breast MRI and without, given especially that the vast majority of those women did not have any pathological finding in their breast MRI examination. Further, this group reflects the typical target cohort of such a technique in application so we believe in a way it represents the target domain quite well.

It is also worth pointing out here that we include around 1000 examinations from the EA1141 trial in our independent testing, which comprises asymptomatic women with dense breast tissue and without additive breast cancer-related risk factors. We hope this helps address the reviewer's concerns.

Regarding generalizability and robustness: We politely disagree with the premise, as we do not believe the training dataset size alone can judge these factors (despite of course influencing them). These factors, in our opinion, are judged on the target domain. For example, organ segmentation typically requires less data than e.g. lesion characterization. In fact, initially we had trained a model with around 25 cases as a proof-of-concept, which seemed to already do the job decently. We rather believe that generalizability and robustness are judged mostly on the evaluation side, as even a model with 1000 training cases can potentially not generalize well if the target domain ends up being too different. This is why we undertook extensive effort on the evaluation side to include >5000 examinations from different centers and manufacturers, as well as undergo manual verification of each case by radiologists.

R3.5. If direct aortic diameter measurement can determine risk, why is a broader risk assessment necessary? What is the clinical significance of risk assessment?

Answer: We thank the reviewer for their comment. Indeed, we just stuck to providing an insight as comprehensive as possible and thus included both the diameter thresholds from the AHA guidelines and the risk scores from the AHA guidelines in our evaluation.

*We assume this makes the data analysis more complete, however if the reviewer and editor opt to remove the AHA guideline risk analysis from the manuscript it can of course be considered. A bit reluctantly, since we think the inclusion can be considered justified and it proves as well a significant point: **With more complex risk analyses being introduced in guidelines the radiologist is faced with ever more complex considerations when analyzing the imaging data.***

*Using an approach as suggested, in term **alleviates all such manual calculation processes from more complex risk models (which might get even more complex in the future)** from the workflow of the radiologists and uses digitalization so that the radiologists can focus on the primary image evaluation. We hope the reviewer agrees to referring as well to such more complex risk stratification scores as already mentioned in the AHA 2022 guidelines (section: 2.3.1. Normalizing Aortic Root and Ascending Aortic Diameters for Body Size; <https://www.ahajournals.org/doi/10.1161/CIR.000000000001106>)*

R3.6. Lack of prospective validation: While the retrospective analysis shows promising results, the true clinical utility and performance of the proposed framework can only be reliably assessed through prospective trials.

Answer: We thank the reviewer for raising this important aspect. Indeed, a prospective cohort has not been part of the study and it is a limitation that we discuss in our limitations section. Whilst we agree that a prospective multicentric randomized clinical trial can be considered gold standard, we are addressing a topic in which the status “disease” is mostly defined by variation of the diameter of an organ - as per the current guidelines.

The target organ itself (aorta) is (inevitably) present in every single patient, thus successfully determining a diameter in the aorta can be seen as mostly allowing to assess the robustness in finding the status “disease” as well.

As we would need probably years of prospective studies with again thousands of new examinations (due to the relatively rare disease) we believe in the benefit of a retrospective approach that is also deployed on data from “new” sites the algorithm has never seen (and to a longitudinal new dataset from our in house examinations). As these sets included different study sites, data from all major vendors and clinical field strengths might be indicative of robustness of finding the aorta, segmenting it and measuring the diameter as well in futurely newly acquired breast MRI examinations.

It is also worth noting that despite its retrospective nature, our dataset from the University Hospital was consecutive, comprising all examinations performed in the premises within a 5 year period, laying a strong foundation that successful prospective evaluation is feasible and justifying the likely costly multi-year endeavor now ahead of us.

*Further, we assume that in a prospective clinical study (in which the radiologists would be aware that in parallel to them an AI is searching for aortic diseases) we **might even potentially introduce a diagnostic awareness bias** limiting real world insights as compared to our evaluation, in which the radiologists reports were usable as real world indicators of reporting frequency of aortic diseases.*

We hope despite the lack of prospective validation in a RCT the reviewer still finds the study to add relevant evidence to the literature.

Minor Concerns:

R3.7. Contrast requirement for segmentation: The manuscript needs to clarify whether the proposed segmentation method is limited to contrast-enhanced breast MRI scans, or if it can also be applied to non-contrast (plain) MRI scans.

Answer: The ANN was trained on contrast enhanced images as it is a pivotal core part of all current routine breast MRI examinations according to the recommendations. With research being focussed as well on non-contrast enhanced breast MRI training the dataset on unenhanced images would be highly interesting, we thank the reviewer for this suggestion. Indeed, given the good morphologic depiction of large vessel structures we would assume the ANN might work in those cases as well. However, in any case the ANN would need to be trained on unenhanced images and we would not expect the current ANN trained on contrast enhanced data not to be affected by the domain shift in case of just deploying it on non-contrast imaging data.

R3.8. How should cases be handled when the ascending or descending aorta measurements exceed the threshold? Is ultrasound test needed?

Answer: Subsequent clarification would likely depend on the individual patient and the size of the aneurysm. Likely ultrasound will play an important role.

Missing details:

R3.9 Please include details about computational resources for development of the model, such as GPU specifications.

Answer: We thank the reviewer for the suggestion. Relevant information is now provided.

R3.10 Addressing these concerns will provide a more comprehensive understanding of the methodology and its potential impact on clinical practice.

We thank the reviewer for the helpful comments. We worked extensively on providing additive analysis, evaluations, and insight into the approach. We hope that we have better highlighted the potential impact on clinical practice.

Review response 2 for manuscript NCOMMS-24-38505-T:

“AI-Enhanced Breast MRI: Unveiling Hidden Aortic Aneurysms through Routine Dual Screening for Cancer and Cardiovascular Health”

We thank the editor and the reviewers for their responses and suggestions, as well as for participating in the review process.

We have carefully considered each point raised and have made the corresponding revisions to the manuscript including

- additive experiments providing a focused spotlight on false positives causing additive burden and costs to a varying degree depending on the clarification pathway and the selected thresholds in the ANN,*
- performing additive experiments in analyzing the ANN performance depending on the training dataset volumes,*
- adding the requested limitations into the discussion section,*
- responding and addressing individually all topics mentioned by each of the reviewers, including the topics mentioned by the added reviewer 4.*

Please find below detailed responses for each comment in italics.

Reviewer 1

This is a resubmission. The authors worked hard to improve the manuscript by addressing all comments. At least for this reviewer the paper improved significantly.

Answer: We thank the reviewer for participating in the review process and for their appreciation of our effort in this revision. Please find responses to the minor comments below.

Some minor comments:

R1.1. Introduction:

Well written but too long. The breast MRI part can be shortened. No one doubts that this is an established method with several indications including screening.

Answer: We slightly shortened the introduction accordingly

R1.2. Stability across different post-GBCA-administration:

Chapter hard to read. Shorten your sentences.

Answer: We thank the reviewer for the suggestion and we would agree. We have now rephrased the chapter.

R1.3. Limitations: retrospective data: May I suggest to add arguments as written in the comments for the reviewers

Answer: We further expanded the limitations section accordingly.

R1.4. Conclusion of discussion: should be rephrased.

Answer: We have now fixed the conclusion.

R1.5. MM: please rephrase: ...Herein, the datasets reflect the diverse settings breast MRI is performed in, in order of strengthening the practical impact of the approach:

Answer: We agree that it was a convoluted sentence and we have simplified it to just "The datasets reflect the diverse settings breast MRI is performed in."

We thank the reviewer for their participation and feedback.

Reviewer 2

During this round of revision, the Authors were able to satisfactorily demonstrate good performance in the few tests, especially aortic diameter quantification and location detection, that were not convincing in the first submission. I particularly appreciate the test on aortic diameter in non-flagged cases, which complements very well the previously-reported results on positive cases and meet the main challenge (rule out false negative).

I still have only minor remarks.

Answer: We thank the reviewer for participating in the review process and for their appreciation of our effort in this revision. Please find responses to the minor remarks below.

R2.1. The rate of repeat measurements that fall within 2mm of each other was good (85.55). However, "The ANN measurements demonstrated a high robustness for different timepoints after GBCA injection" seems quite exaggerated. Indeed, the results (Fig 4a) do not look that good: while the measures are quite centered at a very similar mean, the variability is important. Please, add confidence intervals in figure 4b, second and third lines, and a more balanced discussion of this aspect.

Answer: We now eased the statement suggested and reiterated the numbers to make sure that the result is clear to the reader: "The ANN measurements demonstrated promising consistency for different timepoints after GBCA injection as well as a promising longitudinal consistency of ANN across repeat examinations. However, larger variations were also present."

Regarding the addition of confidence intervals in figure 4b: Adding a confidence interval for the mean would not be feasible as it is rather small (around $\pm 0.5\text{mm}$ from the mean). That is due to

the large number of measurements in our analysis. Instead, we have added prediction intervals, i.e. the range in which a new measurement has 95% probability to fall within. We thank the reviewer for the suggestion.

R2.2. “Significance testing was performed using a t-test with paired samples (AA: $p=0.011/p=0.017$; DA: $p=0.876/p=0.749$; Arch: $p=0.598/p=0.349$, for the Erlangen test set and EA1141, respectively).” Please, reformulate the sentences: neither what was tested nor the meaning of the different p-values are clear.

Answer: We rephrased the text and hope it is more clear now.

R2.3. Please, consider rewriting this sentence: “This workflow integration, which does not require modification of any examination or assessment steps and the ANN therein was found to even demonstrate a slightly lower measurement variation as compared between two radiologists performing >1000 control measurements”.

Answer: We appreciate the pointer – we have split it into two sentences and slightly adapted the text.

R2.4. Please, consider rewriting this sentence: “Herein, the datasets reflect the diverse settings breast MRI is performed in, in order of strengthening the practical impact of the approach”.

Answer: We rephrased the sentence according to your suggestion and simplified it to just “The datasets reflect the diverse settings breast MRI is performed in.”.

We thank the reviewer for their participation and feedback.

Reviewer 3

The updated version addresses most of the previous concerns and shows noticeable improvement. However, a few issues still require attention:

Answer: We thank the reviewer for participating in the review process and for their appreciation of our effort in this revision.

R3.1. While the findings indicate enhanced aneurysm detection rates, the possibility of higher false positive rates, which could lead to unnecessary follow-ups and wasted resources. Should be analyzed through additional experiments.

Answer: As requested we now performed an additive analysis with regards to this issue. We agree on the relevance of the topic, which in in is as well an entire own research field which our study cannot fully explore in a “supplemental style” alone, given the complexity of healthcare economics and the associated considerations (e.g. considering different healthcare systems, reimbursement schemes and patient pathway schemes), the regional variations in handling screening approaches as such (e.g. the variance of screening examinations in breast cancer screening offered in the EU alone) and so on. Still with the additive experiments we now include in the manuscript a focused spotlight on the false positives causing additive burden and costs to a varying degree depending on the clarification pathway and the selected thresholds in the ANN. We include this additive analysis now in the methods, results and discussion section and hope this addresses the question raised by the reviewer.

R3.2. It is important to include in the limitations section that some breast MRI scans may not fully image full regions of the thoracic aorta, such as the aortic arch, which may restrict the method's applicability.

Answer: We thank the reviewer for the suggestion. Indeed, so far we only mentioned it in the Methods and we fully agree that it is important for it to be included in the limitations too. We have amended the text accordingly as requested.

R3.3. An ablation study should be conducted to assess how the size of the training dataset affects model performance.

Answer: We conducted multiple additive ablation studies as requested with different numbers of training samples and the results of this can now be found in Extended data figure E6. In general, the segmentation performance is stable even with a low number of subjects, as previously suggested. We expect the justification for the stability to be related to the reasons we stated at the previous response.

We hope the reviewer agrees with our decision to add the ablation experiments in the extended data with only a brief mention in the main text. We believe generalizability assessments of the independent multi-dataset, multi-site, multi-vendor evaluations that were performed might be the rather relevant experiments conducted. We hope our response satisfies the reviewer.

Addressing these points will further strengthen the manuscript.

We thank the reviewer again for their feedback and participation.

Reviewer 4

While most of the concerns raised by Reviewer 3 have been successfully addressed, I still have a few additional points for consideration, which I have outlined below:

Answer: We thank the reviewer for joining the review process and providing novel feedback.

R4.1. Reviewer 3 raised a valid concern regarding the generalizability and robustness of the model, particularly due to the small-scale sample used in training. While the authors argue that training the model with 25 cases delivered promising performance, I still encourage the authors to conduct more rigorous experiments by varying the number of training samples. This would provide a clearer understanding of the model's generalizability and robustness.

Answer: The results of this could now be found in Extended data figure E6. In general, the segmentation performance is stable even with a low number of subjects, as previously suggested. We expect the justification for the stability to be related to the reasons we stated at the previous response.

Given the results of these additive studies and experiments it might be interpreted that for the amount of training we data utilized we do not have an indication, based on the ablation study, that it was chosen insufficiently small (and it might be even hypothesized that we might have potentially saved some tedious annotation time).

We hope the reviewer agrees with our decision to add the ablation experiments in the extended data with only a brief mention in the main text. We believe generalizability assessments of the

independent multi-dataset, multi-site, multi-vendor evaluations that were performed might be the rather relevant experiments conducted in this regard. We hope our response satisfies the reviewer.

R4.2. The authors use the nnU-Net architecture for segmentation, which is a strong choice, but the manuscript lacks sufficient detail regarding the hyperparameter tuning process. Specifically, it is unclear which hyperparameters were tunable, how they were optimized (e.g., learning rate, batch size), and how the model was optimized for this particular task. Given that nnU-Net is highly adaptable, understanding the specific configuration choices made for this task is critical for reproducibility and future model development.

Answer: We agree with the reviewer that these are relevant aspects and we politely point out to the reviewer that these requested hyperparameters were already present in the manuscript (please see “Training of the ANN” section of “Methods” reading “We utilized the high image resolution 3D configuration, with z-score normalization, 1000 epochs, 250 iterations per epoch, 50 validation iterations per epoch, initial learning rate 0.01, weight decay 3×10^{-5} , batch size 2, and patch size $56 \times 192 \times 224$ ”). These parameters were automatically determined by nnU-Net’s self configuring process and we slightly amended the manuscript to reflect that – we did not employ a manual tuning process. The only adaptation by us was the augmentation scheme, which we also have detailed in this section.

We thus hope the level of detail presented is sufficient for the reviewer and are happy to provide any additive details if requested.

R4.3. Statistical tests are extensively used throughout the evaluation. However, multiple statistical tests are performed across different datasets (Erlangen, Duke, EA1141) and analyses (e.g., segmentation performance, aortic diameter measurements, risk assessment). The manuscript does not explicitly mention whether any corrections for multiple hypothesis testing were applied. Given the potential for inflated Type I error due to multiple comparisons, it is important that the authors address this and provide clarity on whether such corrections were applied, or justify why they were not deemed necessary.

Answer: We thank the reviewer for inquiring about the statistical analysis. The three datasets used are completely independent and we would not consider it appropriate to correct across them.

Regarding the analyses, while they are of course related to a small extent, we are still looking to examine different aspects of the model performance rather than e.g. looking for 10 different diseases in the data. Due to this, while indeed Type I errors are always possible (and of course represented by the standard 0.05 threshold), we chose to stick to standard p thresholds and a more straightforward analysis. It is worth stating here that this decision is not motivated by any desire to preserve good results, as any reasonable p threshold (e.g. even 0.05/3) would not have impacted the conclusions of the manuscript given the numbers already present. Nonetheless, if there is a strong argument against this thought process, we could consider implementing testing corrections and we appreciate the comment.

R4.4. While the authors mention performing data augmentation during training (e.g., Rician noise, motion artifacts, and MRI spike artifacts), the manuscript does not explicitly evaluate the model's performance when subjected to noise or artifacts during inference. A dedicated evaluation of the model's robustness to different types and levels of noise or perturbations is necessary to ensure its reliability in clinical practice. This would include testing the model on noisy or perturbed MRI data, which is often encountered in routine clinical settings.

Answer: We thank the reviewer for their comment. Indeed, given the diversity of the included MR dataset from various vendors, acquisition protocols and sites, it is worth pointing out that lots of "natural" noise/artifacts were definitely present in the data, especially in the public datasets. Artificial noise also comes with its own set of problems (e.g. concerns might be raised about the realism of the utilized synthetic data and in general we would rather have a large study with multiple datasets as we do now, rather than testing on artificial, potentially non-representative noise). We however agree that this could be a part of our limitations and we have now added it in the discussion accordingly.

R4.5. The manuscript acknowledges that breast MRI scans were not ECG-gated, which could introduce variability in aortic diameter measurements due to cardiac motion. While the authors note that they accounted for a 2mm variance threshold, this may still affect the accuracy of measurements, especially for smaller aneurysms or those near the diagnostic threshold. It would be beneficial for the authors to discuss the potential limitations of non-ECG-gated imaging in more detail. Could the inclusion of ECG-gated sequences improve measurement accuracy? Additionally, it would be valuable to explore how the ANN might perform when applied to ECG-gated MRI images and whether future work could include this enhancement.

Answer: We agree with the reviewer and we have expanded the discussion accordingly, by introducing an entire paragraph on the topic.

R4.6. There is limited discussion on the interpretability of the model's decision-making process. In clinical applications, understanding the rationale behind a model's predictions is crucial for radiologists and clinicians. The authors should consider incorporating explainability techniques to improve the transparency of the model's decision-making. This would help clinicians understand which aspects of the MRI images contribute to the model's findings and increase confidence in its outputs.

Answer: We appreciate the comment and we agree that interpretability is indeed an important aspect in medical AI. This is why we chose to present to the radiologists not only the final decision, but also the images including the AI derived segmentation masks on the original anatomical structure and the corresponding diameter measured by the algorithm - so that full transparency is achieved in the end-to-end approach. We want to further clarify that technically there is no model decision-making process present in the manuscript. As such, no additive clinical interpretability would make sense as the AI component itself is a segmentation model. Any patient classification is directly based on the calculated diameter in 3D space. It is also worth mentioning that interpretability would be high in any clinical application of this approach, as a physician will be able to directly see the images of the aorta in the medical report alongside the measured diameter.

We thank the reviewer again for their participation

Review response #3 for manuscript NCOMMS-24-38505:

“Unveiling Hidden Aortic Aneurysms through AI-Enhanced Cardiovascular Screening in Routine Breast MRI”

Reviewer 3

Previous comments were mostly addressed.

Answer: We thank the reviewer for participating in the review process.

Reviewer 4

While the authors have satisfactorily addressed most of my concerns, a few minor issues remain that I would like to see clarified:

Answer: We thank the reviewer for participating in the review process.

1. Hyperparameters: Given the critical role of hyperparameters in achieving optimal performance in ANNs generally, please elaborate on how nnU-Net determines these values. Are they the result of an optimization process, or are they default settings derived from prior experiments? A brief explanation would be especially useful for readers who are not familiar with the nnU-Net framework.

Answer: Thank you, as requested, in the following you can find an overview of all the hyperparameters mentioned already in our manuscript regarding nnU-Net hyperparameters (as to be found as well here: <https://www.nature.com/articles/s41592-020-01008-z/figures/2>).

We further added more mentions of hyperparameters from the nnU-Net directly in our manuscript (in the section “Training of the ANN”), we hope this addresses the question raised. Here follows a description of all hyperparameters:

Empirically set hyperparameters (set by the nnU-Net developers and do not depend on data):

- *Initial learning rate: 0.01 [already mentioned]*
- *loss selection: combined loss, summation of dice and binary cross-entropy losses [already mentioned]*
- *architecture: U-Net architecture with plain convolutions for feature extraction [was present in initial submission, then removed, now re-added]*
- *optimizer: stochastic gradient descend [standard approach, but now also described in the manuscript]*

- *data augmentations: defaults + custom extensions by us [already described in the manuscript]*
- *training epochs / iterations per epoch: 1000 epochs, 250 iterations/epoch [already mentioned]*

Rule-based hyperparameters (i.e., derived from the data):

The applicable rules depend on: data size, spacing, being CT or not, and whether the data are anisotropic. While this information was present in the manuscript indirectly, we now added explicit description of them in the “Training of the ANN” section. Specifically, these are the hyperparameters:

- *Intensity normalization: z-score [already mentioned; only CT data have other normalization scheme]*
- *Image resampling strategy: We have anisotropic data → third-order spline, out-of-plane with nearest neighbor [now included in manuscript]*
- *Annotation resampling strategy: We have anisotropic data → in-plane with linear interpolation, out-of-plane with nearest neighbor [now included in manuscript]*
- *Image target spacing: We preprocess the data to spacing “0.85x0.85x1.5 mm” (as described in “Methods > Data preparation > Preprocessing”), so this is what nnU-Net uses directly without the need to derive. We used “0.85x0.85x1.5 mm” because almost all our training data were already at this spacing.*
- *patch size / batch size: [already mentioned]*
- *The rest are not applicable since they are not utilized (e.g. low-resolution models)*

2. Model’s Decision-Making Process: To clarify my previous comment, I am referring specifically to how the model arrives at its segmentation predictions. Although explainability techniques in image classification have received considerable attention, similar methods for semantic segmentation have been relatively neglected. A discussion on this topic, potentially discussing recent surveys such as [1], would enhance the paper.

[1]: Gipiškis, Rokas, Chun-Wei Tsai, and Olga Kurasova. "Explainable AI (XAI) in image segmentation in medicine, industry, and beyond: A survey." *ICT Express* 2024.

Answer: We have now expanded the discussion to list the lack of segmentation explainability methods as a limitation, citing the mentioned study.